# Bile acids-mediated intracellular cholesterol transport promotes intestinal cholesterol absorption and NPC1L1 recycling

Jian Xiao [1,5], Le-Wei Dong[1,5], Shuai Liu[1,2], Fan-Hua Meng[1,2,3], Chang Xie[1], Xiao-Yi Lu[1], Weiping J. Zhang [4], Jie Luo [1] & Bao-Liang Song [1] ✉

Niemann-Pick C1-like 1 (NPC1L1) is essential for intestinal cholesterol absorption. Together with the cholesterol-rich and Flotillin-positive membrane microdomain, NPC1L1 is internalized via clathrin-mediated endocytosis and transported to endocytic recycling compartment (ERC). When ERC cholesterol level decreases, NPC1L1 interacts with LIMA1 and moves back to plasma membrane. However, how cholesterol leaves ERC is unknown. Here, we find that, in male mice, intracellular bile acids facilitate cholesterol transport to other organelles, such as endoplasmic reticulum, in a non-micellar fashion. When cholesterol level in ERC is decreased by bile acids, the NPC1L1 carboxyl terminus that previously interacts with the cholesterol-rich membranes via the $A_{1272}LAL$ residues dissociates from membrane, exposing the $Q_{1277}KR$ motif for LIMA1 recruitment. Then NPC1L1 moves back to plasma membrane. This study demonstrates an intracellular cholesterol transport function of bile acids and explains how the substantial amount of cholesterol in NPC1L1-positive compartments is unloaded in enterocytes during cholesterol absorption.

Cholesterol is an important lipid that regulates membrane properties and modifies proteins including Hedgehog and Smoothened[1–3]. A high level of cholesterol is the key risk factor for cardiovascular disease (CVD) and nonalcoholic fatty liver disease[4,5]. Cholesterol can be obtained from de novo biosynthesis and dietary absorption. Higher consumption of dietary cholesterol is significantly associated with higher risk of CVD[6,7].

Niemann-Pick C1 like 1 (NPC1L1) is a 13-transmembrane protein which is absolutely required for intestinal cholesterol absorption[8,9]. Our previous studies have shown that NPC1L1 takes up dietary cholesterol through vesicular endocytosis[10,11]. The binding of cholesterol to the amino terminal domain of NPC1L1 induces the dissociation of the carboxyl terminus from the plasma membrane (PM), exposing the endocytic motif $Y_{1306}VNxxF$ (where x stands for any amino acids) for NUMB recognition[12,13]. NUMB recruits AP2/clathrin to generate

clathrin-coated vesicles and initiates NPC1L1 endocytosis. Meanwhile, NPC1L1 constitutively interacts with Flotillin-1/−2 to form cholesterol-rich membrane microdomains[14]. In such a vesicular trafficking way, NPC1L1 transports a large amount of cholesterol to the endocytic recycling compartment (ERC), which is a Rab11-positive endosome enriched in cholesterol and considered as an intracellular cholesterol pool[15]. When the cholesterol level in ERC drops, NPC1L1 interacts with LIM domain and actin binding 1 (LIMA1) via its $Q_{1277}KR$ residues and is recycled back to the PM by LIMA1 and the associated myosin Vb[16]. Mutation of *LIMA1* causes intracellular retention of NPC1L1 and decreases intestinal cholesterol absorption[16]. Ezetimibe blocks the cholesterol absorption by binding to NPC1L1 and inhibiting its internalization[11,14,17,18].

The cholesterol in ERC is transported to the endoplasm reticulum (ER) where acyl-CoA: cholesterol acyltransferase 2 (ACAT2) converts

[1]College of Life Sciences, Taikang Center for Life and Medical Sciences, Taikang Medical School, Hubei Key Laboratory of Cell Homeostasis, Wuhan University, Wuhan, China. [2]Heart Center, First Affiliated Hospital of Xinjiang Medical University, Urumqi 830054 Xinjiang, China. [3]Affiliated Hospital of Jining Medical University, Jining 272007 Shandong, China. [4]Department of Pathophysiology, Naval Medical University, Shanghai, China. [5]These authors contributed equally: Jian Xiao, Le-Wei Dong. ✉e-mail: blsong@whu.edu.cn

cholesterol to cholesteryl esters (CEs), which are then incorporated into chylomicrons and secreted to lymph. However, the immunohistochemical staining results showed that the Rab11-positive ERC displayed little overlap with the ACAT2-positive ER in intestinal enterocytes[11]. It is unknown how the ERC cholesterol taken up by NPC1L1 is transported to the ER. Here, we find the four specific bile acid (BA) species−chenodeoxycholic acid (CDCA), deoxycholic acid (DCA), taurodeoxycholic acid (TDCA) and glycodeoxycholic acid (GDCA)− can transport ERC cholesterol to the ER and PM at the submicellar concentration (0.1 mM-1 mM). The reduction of cholesterol in NPC1L1-positive vesicles exposes $Q_{1277}KR$ to LIMA1, an event that is prevented by cholesterol-regulated binding of NPC1L1 $A_{1272}LAL$ residues to vesicle membrane, and induces NPC1L1 translocation to the PM. By decreasing ERC cholesterol, BAs favor NPC1L1 recycling back to the cell surface and increase cholesterol absorption.

## Results

### NPC1L1 endocytosis is required for cholesterol absorption

To confirm that NPC1L1 endocytosis is required for cholesterol absorption, we generated the *Npc1l1*-$Y_{1306}$VNYGF to AAAYGA (4A) knock-in mutation mice that harbor the impaired endocytic sequence[13] (Fig. 1a). Cholesterol was taken into the enterocytes of wild-type (WT) mice but retained in the brush border membrane of homozygous 4A mice after cholesterol gavage (Fig. 1b). The uptake of $^3$H-cholesterol by the livers of heterozygous and homozygous 4A mice were lowered by ~35% and ~60% relative to WT mice, respectively (Fig. 1c). The amount of plasma $^3$H-cholesterol in the heterozygotes and homozygotes were reduced by ~29% and 53% (Fig. 1d). Together, these results confirm the previous findings that endocytosis of NPC1L1 is essential for intestinal cholesterol absorption[13,16].

### Bile acids promote NPC1L1 translocation to the PM

CRL1601/NPC1L1-3×Myc-EGFP is a rat hepatic cell line stably expressing NPC1L1-3×Myc-EGFP[14]. The Myc-tag epitope is present in the extracellular/luminal side and can be recognized by the anti-Myc antibody when NPC1L1 is on the PM (Fig. 2a, S1a). In the cells grown under normal conditions (10% FBS medium), a majority of NPC1L1 was found in the ERC, where cholesterol was enriched (Fig. 2b, Fig. S1a). Consistent with previous results[10,19], cyclodextrin (CDX) depleted cholesterol from the ERC and induced NPC1L1 relocalization to the PM (Fig. S1a). To explore whether BAs play a role downstream of NPC1L1, we treated the CRL1601/NPC1L1-3×Myc-EGFP cells with mouse bile− untreated, boiled, or dialyzed−and *Sus scrofa* derived BA mixture. Cells exposed to bile, boiled bile or BA mixture had about 1 mM total BAs in the medium, whereas those to dialyzed bile had very little BAs (Fig. 2b, Figure S2a). Similar to CDX, bile caused NPC1L1 to transport to the PM from the ERC (Fig. 2b). Meanwhile, the cholesterol in ERC was decreased and dispersed throughout the cell (Fig. 2b). The same phenotypes were observed in the cells treated with the boiled but not dialyzed bile (Fig. 2b). These results suggest that small molecules, instead of proteins, decrease the ERC cholesterol and induce PM localization of NPC1L1. The BA mixture could similarly induce PM localization of NPC1L1 as well as a decrease of ERC cholesterol (Fig. 2b−d and S2c).

Among 12 main BA species (Fig. S2b), CDCA, DCA, TDCA and GDCA showed strong potency to induce the PM localization of NPC1L1 and cholesterol dispersal. Other BA species, including cholic acid (CA), taurocholic acid (TCA), glycocholic acid (GCA), taurochenodeoxycholic acid (TCDCA), glycochenodeoxycholic acid (GCDCA), lithocholic acid (LCA), glycolithocholic acid (GLCA), and taurolithocholic acid (TLCA), had little effect on NPC1L1 localization and cholesterol distribution in the cell (Fig. 2b−d and Fig. S2c−g). BA mixture and the four BA species (CDCA, DCA, GDCA and TDCA) transport cholesterol away from the Rab11a or TFR-positive ERC without causing ERC redistribution (Fig. S3a, b). BA mixture

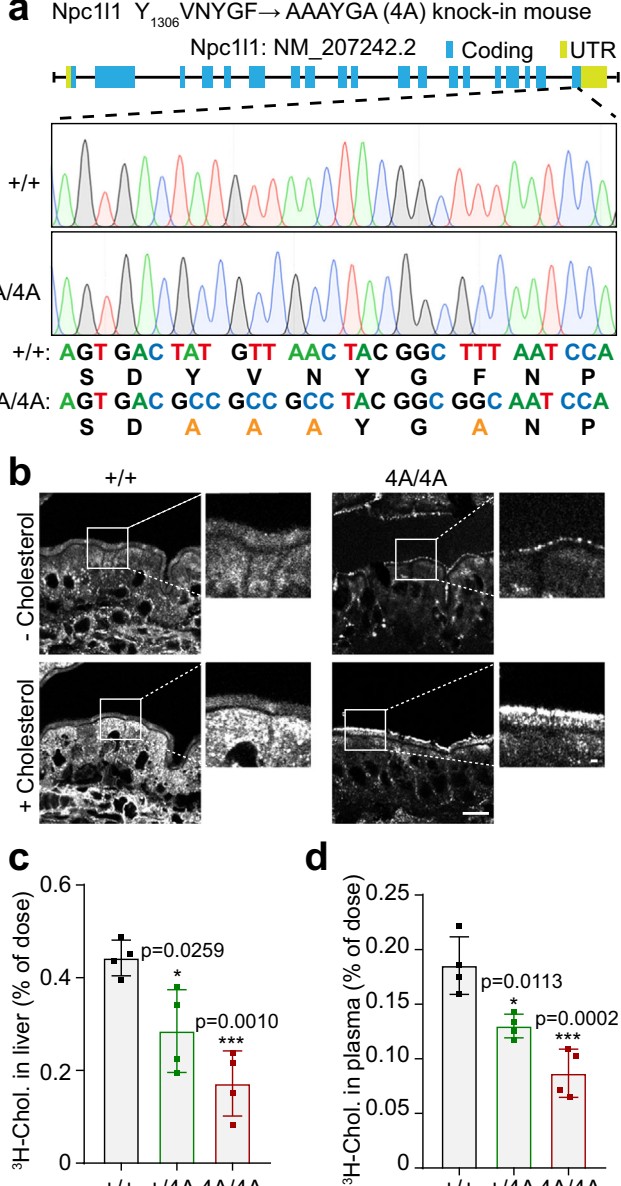

**a** Npc1l1  $Y_{1306}$VNYGF→ AAAYGA (4A) knock-in mouse

Npc1l1: NM_207242.2    ■ Coding  ■ UTR

+/+: AGT GAC TAT GTT AAC TAC GGC TTT AAT CCA
     S   D   Y   V   N   Y   G   F   N   P

4A/4A: AGT GAC GCC GCC GCC TAC GGC GGC AAT CCA
       S   D   A   A   A   Y   G   A   N   P

**b**

− Cholesterol    +/+    4A/4A

+ Cholesterol    +/+    4A/4A

**Fig. 1 | NPC1L1 endocytosis is essential for cholesterol absorption. a** The schematic model of *Npc1l1*-$Y_{1306}$VNYGF → AAAYGA (4A) knock-in mouse. +/+: Wild-type (WT); UTR: Untranslated region. **b** Filipin staining of intestinal sections from WT male mice (+/+) or 4A knock-in male mice (4A/4A) receiving an oral gavage of 30 mg/mL cholesterol in 100 μL corn oil (+ Cholesterol group) or 100 μL corn oil (- Cholesterol group). Scale bar, 10 μm (main); 1 μm (inset). The relative level of $^3$H-cholesterol in the liver (**c**) and plasma (**d**) in the WT (+/+), heterozygous (+/4A) or homozygous (4A/4A) 4A knock-in male mice. Chol, cholesterol. Values were presented as mean ± SD (*n* = 4 mice). One-way ANOVA with Tukey post hoc test, \**P* < 0.5; \*\**P* < 0.01. Source data are provided as a Source Data file.

redistributed ERC cholesterol but did not alter PI(4,5)$P_2$ or PI3P distribution (Fig. S3c, d).

To test whether BAs affect NPC1L1 and cholesterol distribution in the enterocytes, we utilized a knock-in mouse line in which the FLAG tag and the EGFP tag were sequentially fused at the carboxyl terminus of NPC1L1 (Fig. S1b). We isolated intestine sections from the neonatal *Npc1l1-EGFP* knock-in mice for in vitro culture as previously described[20] (Fig. 2e). NPC1L1 mainly resided on the brush border in the cholesterol-free medium (- Cholesterol group in Fig. 2e). Cholesterol gavage caused the internalization of NPC1L1 and cholesterol, which

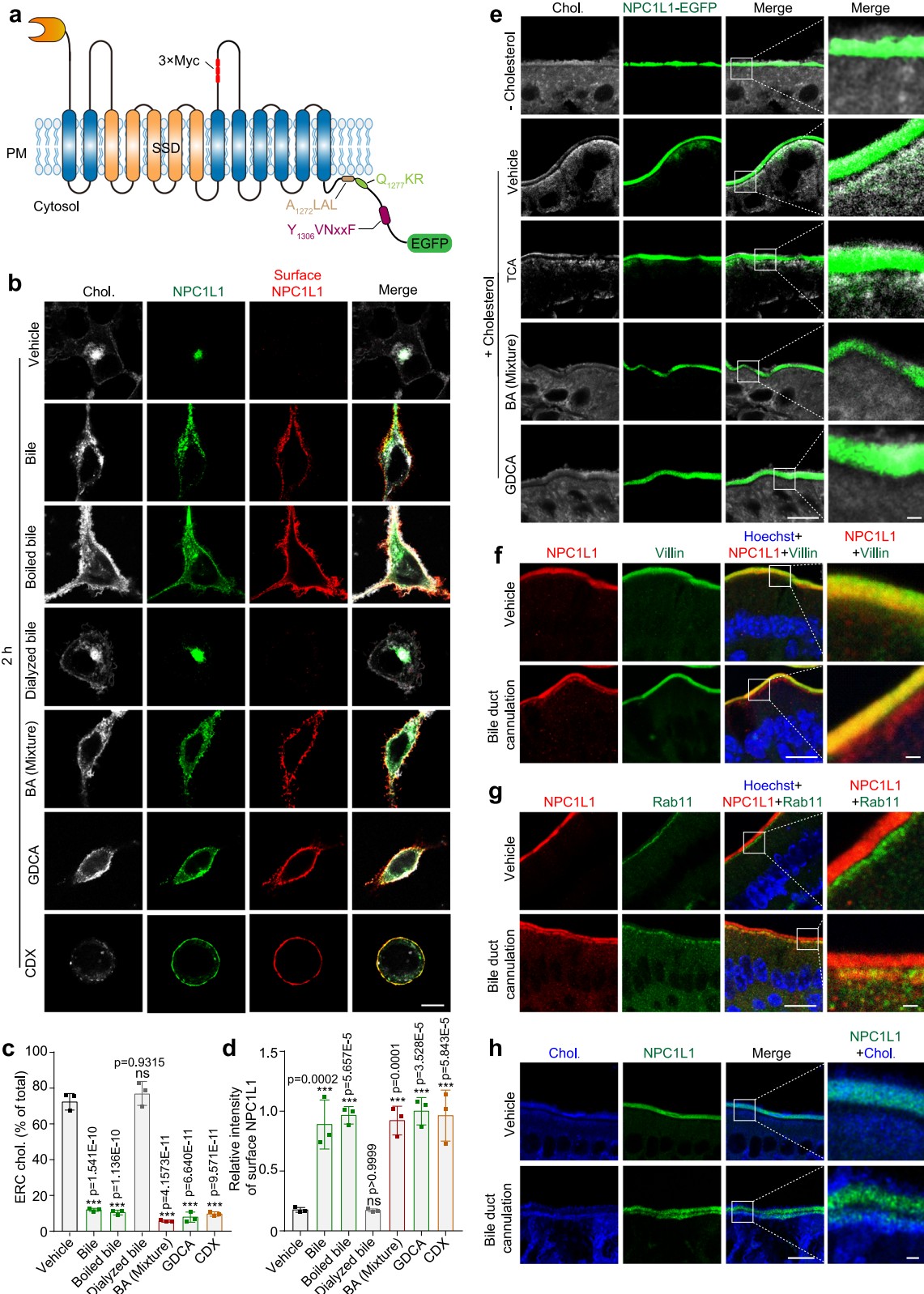

appeared as a layer below the brush border membrane (vehicle subgroup in Fig. 2e). The same pattern was observed in the group co-treated with TCA, since TCA did not mobilize intracellular cholesterol or cause NPC1L1 relocation to the PM (TCA subgroup in Fig. 2e). On the contrary, the internalized cholesterol and NPC1L1 were not seen in the intestine of mice treated with BA mixture or GDCA, both of which can mobilize ERC cholesterol and therefore cause NPC1L1 transport to the

PM (BA mixture and GDCA subgroup in Fig. 2e). We next deprived bile from mouse intestine using a surgical cannulation of common bile duct. Immunofluorescent staining showed that a portion of NPC1L1 remained underneath the apical membrane of enterocytes of cannu-lated mice (Fig. 2f). The intracellular NPC1L1 colocalized with Rab11 (Fig. 2g), as seen in cultured cells and mouse intestine[10,11]. By contrast, the NPC1L1 protein in vehicle-treated mice was restricted exclusively to

**Fig. 2 | BAs promote NPC1L1 transportation to PM. a** The topology of human NPC1L1-3×Myc-EGFP. The 3×Myc tag was inserted after the 966th amino acid of NPC1L1 and faced the extracellular/luminal side. $A_{1272}LAL$, $Q_{1277}KR$ and $Y_{1306}VNxxF$ were cytosolic amino acids of NPC1L1. The EGFP protein was fused with the carboxyl terminus of NPC1L1. SSD: sterol-sensing domain. **b** CRL1601/NPC1L1-3×Myc-EGFP cells were treated with bile, dialyzed bile, boiled bile, BA mixture, GDCA or CDX for 2 h. The surface NPC1L1 was labeled by anti-Myc antibody prior to permeabilization, and cellular cholesterol was stained by filipin. Scale bar, 10 μm. Percentages of ERC cholesterol relative to total cholesterol (**c**) and relative intensity of surface NPC1L1 (**d**) in (**b**) were quantified. The ERC cholesterol was defined as the cholesterol colocalized with intracellular NPC1L1. The average intensity of surface NPC1L1 in GDCA-treated cells was defined as 1. Values were presented as mean ± SD (n = 3 independent trials, 100 cells/trial). One-way ANOVA with Tukey post hoc test, \*\*\*$P < 0.001$; ns, no significance. **e** Filipin staining of in vitro cultured intestine sections from *Npc1l1-EGFP* knock-in mice. Small intestine from neonatal mouse was incubated with or without 10 μg/mL cholesterol in the presence or absence of indicated BA species or BA mixture for 2 h, and then fixed by 4% PFA (n = 2). Chol, cholesterol. Boxed areas are shown at a higher magnification on the right. Scale bar, 10 μm (main); 1 μm (inset). Deparaffinized sections of intestinal samples from mice receiving sham surgery or bile duct cannulation were stained with anti-NPC1L1 and anti-Villin (**f**), or with anti-Rab11 antibodies (**g**), followed by counterstaining with Hoechst (n = 2). Boxed areas are shown at a higher magnification on the right. Scale bar, 10 μm (main); 1 μm (inset). **h** Filipin staining of intestinal sections from *Npc1l1*-EGFP knock-in mice receiving sham surgery or bile duct cannulation (n = 2). The control group received sham surgery. Chol, cholesterol. Boxed areas are shown at a higher magnification on the right. Scale bar, 10 μm (main); 1 μm (inset). Chol: Cholesterol. Source data are provided as a Source Data file.

villin-positive brush border membrane (Fig. 2f). In the mice with bile duct cannulation, a portion of NPC1L1-EGFP protein distributed beneath the brush border membrane and stained positive for cholesterol (Fig. 2h). Based on these data and our previous results[2], we hypothesized that BAs cause cholesterol depletion in Rab11-positive ERC and then NPC1L1 is transported to the PM from the ERC.

## BAs mobilize cholesterol from the ERC to the ER and PM

BAs are reported to promote cholesterol transport in vitro and in cultured cells at a submicellar concentration (0.15-0.6 mM)[21–23]. Submicellar BAs can extract cholesterol from the membrane by mitigating the interaction between cholesterol and sphingomyelin, and then promote nondirectional cholesterol traffic among membranes[23]. To verify BAs below their critical micellar concentrations (CMC) could still convey cholesterol, we applied all 12 BAs to the in vitro cholesterol mobilizing assay as depicted in Fig. 3a. Artificial liposomes with or without cholesterol were placed in phosphate buffered saline (PBS)-filled chambers separated by a semi-permeable membrane, which allows for free diffusion of molecules below 50 kDa. Saline, BA mixture or different BA species were pipetted into cholesterol-liposome side, and the amounts of cholesterol at the opposite side were measured. We found that all dihydroxy BAs at 0.1 mM concentration could mobilize cholesterol, while the other BAs failed to do so (Fig. 3b). In line with the previous findings[21,22], GCDCA and TCDCA at 0.1 mM concentration had less potency on cholesterol transport than CDCA, DCA, GDCA and TDCA (Fig. 3b), which might account for the cholesterol entrapment in the ERC of GCDCA or TCDCA-treated cells (Fig. S2d). The six BAs (CDCA, DCA, GDCA, TDCA, GCDCA and TCDCA) transported cholesterol in a concentration-dependent manner in vitro (Fig. S4a). The highest concentration (1 mM) of BAs used in this experiment is still below their CMC. TCA did not mediate cholesterol transport (Fig. S4a). In addition, the dynamic light scattering assay showed that the liposomes remained intact until the BA concentration increased to 10 mM (Fig. S4b). These data indicate that the dihydroxy BAs promote cholesterol transfer directly at their submicellar concentrations without disrupting liposome structure.

We further assessed whether BAs could mobilize cholesterol from the ERC to other organelles, such as the ER and the PM. ACAT1 and ACAT2 convert cholesterol to CEs in the ER for incorporation into chylomicrons and secretion into lymph. The amount of nascent CEs is an index for cholesterol reaching ER. The BA mixture and four specific BA species (CDCA, DCA, GDCA and TDCA) increased CE levels by about two folds (Fig. 3c), and the increment was completely abolished by the ACATs inhibitor avasimibe. Consistently, the formation of lipid droplets in CRL1601 cells and in vitro cultured intestinal enterocytes was increased by these BAs but blunted by avasimibe (Fig. 3d–f). All of these nascent CEs colocalized well with the cytosolic lipid droplet marker Plin2 in CRL1601 cells[24] (Fig. S4c). But only a portion of CEs induced by BA mixture were colocalized with Plin2-positive lipid droplets in the in vitro cultured intestine (Fig. S4d). It is possible that the

Plin2-negative lipid droplets were in the ER for lipoprotein secretion. Since high concentration of ER cholesterol impedes SREBP2 cleavage, we analyzed the SREBP2 processing as an index for cholesterol arriving at the ER. Cholesterol starvation did not alter ERC cholesterol distribution but increased the nuclear form of SREBP2 (Fig. S4e, f). The BA mixture and the four specific BA species (CDCA, DCA, GDCA and TDCA) effectively inhibited SREBP2 processing (Fig. S4f), suggesting that they mobilize cholesterol to the ER from the ERC.

To assess the level of cholesterol in the PM, CRL1601/NPC1L1-3×Myc-EGFP cells were incubated with purified mCherry-D4H protein, a widely used cholesterol probe[25], for 30 min on ice without cell permeabilization. The four BA species (CDCA, DCA, GDCA and TDCA) and BA mixture caused remarkable increase of PM cholesterol (Fig. 3g, h). Inhibition of cholesterol esterification by avasimibe augmented the effects of four BA species (CDCA, DCA, GDCA and TDCA) or BA mixture on PM cholesterol (Figure S4g, h). TCA that barely mobilized cholesterol (Fig. 3b) failed to increase the formation of CEs and lipid droplets, inhibit SREBP2 cleavage, or increase PM cholesterol level (Fig. 3 and S4f–h).

Certain BAs including CDCA are the ligands of farnesoid X receptor (FXR)[26]. To rule out the possibility that FXR was involved in the BA-regulated transport of cholesterol and NPC1L1, we treated the cells with the FXR agonist GW4064. GW4064 and CDCA, but not BA, upregulated the mRNA of *Shp*, which is a downstream gene induced by FXR[27] (Fig. S5a). GW4064 barely facilitated the translocation of NPC1L1 to PM and cholesterol egress from the ERC compared with CDCA and BA (Fig. S5b–d). Altogether, these results suggest that BAs carry cholesterol out of the ERC and deliver cholesterol to the ER by solubilizing cholesterol.

## BAs function within the cell

The Na⁺-taurocholate transporting polypeptide (NTCP) is a PM-resident protein and the major transporter for sodium bile salts in rat liver cells[28]. We next investigated the role of NTCP in BA-mediated cholesterol transportation. The expression of *Ntcp* in CRL1601/NPC1L1-3×Myc-EGFP cells was knocked down (KD) by small interfering RNAs (siRNAs) against *Ntcp* (Fig. S6a). Silencing of *Ntcp* decreased cellular levels of GDCA and TDCA, but not those of CDCA and DCA (Fig. S6b). These results are consistent with the previous findings that NTCP preferentially shuttles glycine or taurine conjugated bile salts but not the unconjugated ones[29,30].

GDCA and TDCA could not enter the cells (Fig. S6b) and failed to transport cholesterol out of the ERC or redistribute NPC1L1 to the PM in *Ntcp* silenced cells (Fig. 4a–c). In contrast, CDCA and DCA could still enter the cells, diffuse ERC cholesterol and promote NPC1L1 translocation to PM in the *Ntcp* knockdown cells (Fig. S6c–e), since these BA species can be transported into cells by other transporters[31]. Consistently, NTCP ablation blocked the formation of CEs and lipid droplets induced by GDCA or TDCA, but not by CDCA or DCA (Fig. 4d–f and S6f–h). Knockdown of *Ntcp* blocked the GDCA and TDCA but not

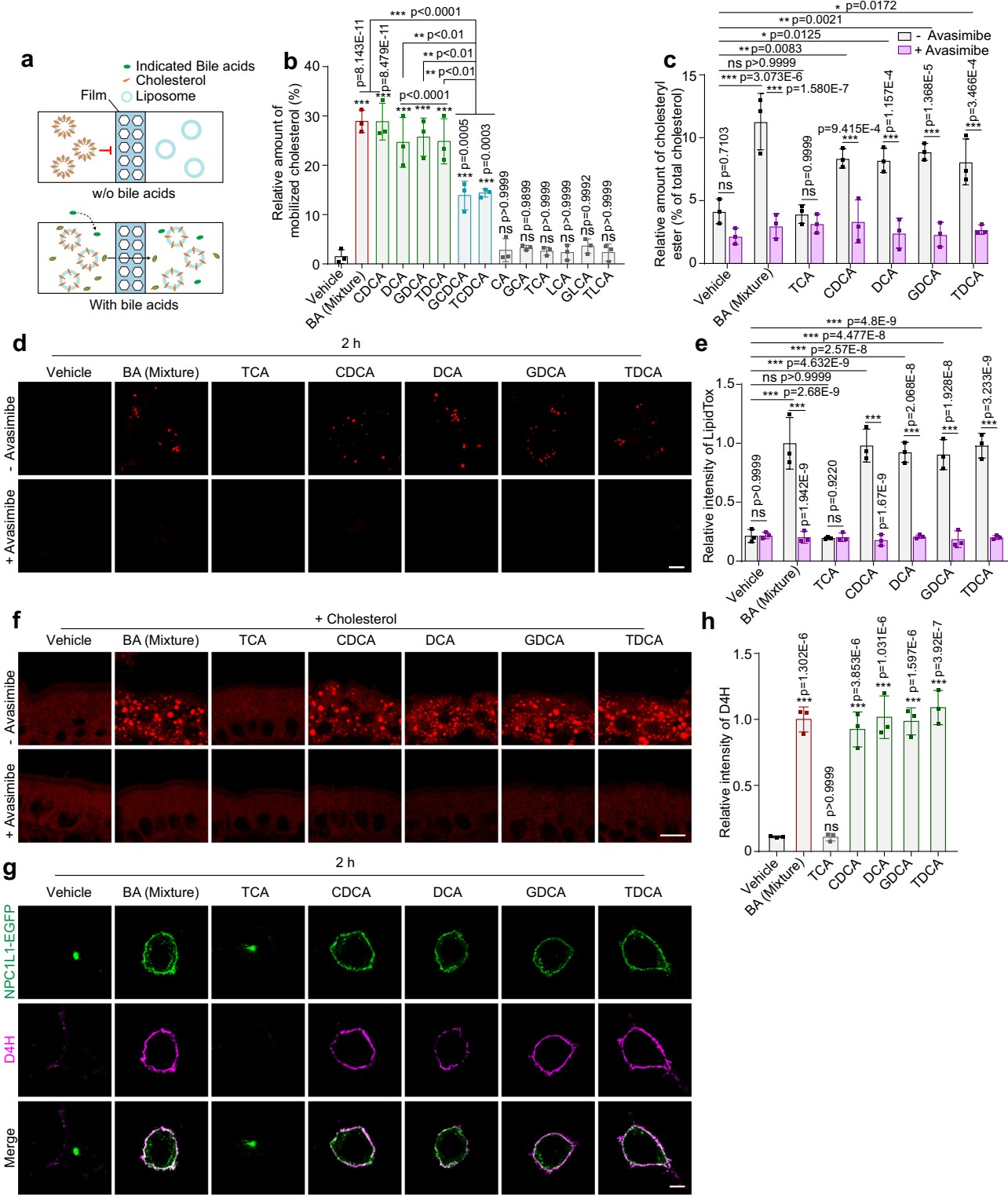

CDCA or DCA-induced increase in PM cholesterol (Fig. 4g, h and Fig. S6i, j). Collectively, these results suggest that BAs are transported into the cells, mobilize the ERC-cholesterol and then induce NPC1L1 translocation.

BAs can enter the enterocytes via passive diffusion or apical sodium-dependent bile acid transporter (ASBT). Passive diffusion mediates the entry of unconjugated BAs in the jejunum[32], even the passive permeability of CDCA in the jejunum is 9-fold that in the ileum[33]. ASBT is primarily in the ileum (Fig. S7a) and almost completely resorbs conjugated BAs from intestine. To explore the role of ASBT in

ileal cholesterol absorption, we treated mice with oral administration of linerixibat, a selective ASBT inhibitor[34]. Linerixibat treatment increased BA excretion by ~2-fold in feces (Fig. S7b). Administering mice with linerixibat lowered total cholesterol in serum by ~26% and increased cholesterol excretion into feces by ~42% than the vehicle-treated mice (Fig. S7c, d), while the total triglyceride in serum and feces remained unaltered in the linerixibat-treated group (Figure S7e, f). In addition, linerixibat gavage reduced cholesterol absorption by ~43% than the vehicle group (Fig. 4i). Immunohistochemical staining revealed a sequestration of NPC1L1 and cholesterol beneath the ileal

**Fig. 3 | BAs are intrinsic cholesterol mobilizers that convey cholesterol from the ERC to the ER and PM. a** In vitro cholesterol mobilization assay. Artificial liposomes with or without cholesterol were separated by a semi-permeable membrane. After addition of indicated bile acids into the cholesterol-liposome side, the liposomes from the other side were harvested and subjected to cholesterol determination. **b** Percentages showing cholesterol mobilized by the indicated BAs. The value of total cholesterol mobilized to the non-cholesterol liposome was divided by that of total cholesterol in the cholesterol-liposome. Data were presented as mean ± SD ($n = 3$). One-way ANOVA with Tukey post hoc test, ***$P < 0.001$; ns, no significance. **c** Percentages of cholesteryl esters relative to total cholesterol in CRL1601/NPC1L1-3×Myc-EGFP cells with a 2-h incubation of indicated BAs in the presence or absence of avasimibe. Values were presented as mean ± SD ($n = 3$). Two-way ANOVA with Tukey post hoc test, *$P < 0.05$; **$P < 0.01$; ***$P < 0.001$; ns, no significance. **d** Representative LipidTox staining of CRL1601/NPC1L1-3×Myc-EGFP cells

after 2-h incubation of indicated BAs in the presence or absence of avasimibe. Scale bar, 10 μm. **e** Relative intensity of LipidTox staining in (**d**) was quantified. The average intensity of BA mixture-treated cells in the non-avasimibe group was defined as 1. Values were presented as mean ± SD ($n = 3$ independent trials, 100 cells/trial). Two-way ANOVA with Tukey post hoc test, ***$P < 0.001$; ns, no significance. **f** LipidTOX staining of in vitro cultured intestine sections from mice ($n = 2$). Small intestines from neonatal mice were incubated with 10 μg/mL cholesterol and indicated BAs for 2 h, then fixed by 4% PFA. Scale bar, 10 μm. **g** D4H staining of CRL1601/NPC1L1-3×Myc-EGFP cells with 2-h indicated BA treatments. Scale bar, 10 μm. **h** The relative intensity of D4H in (**g**) was quantified. The average intensity of D4H in BA mixture-treated cells was defined as 1. Values were presented as mean ± SD ($n = 3$ independent trials, 100 cells/trial). One-way ANOVA with Tukey post hoc test, ***$P < 0.001$; ns, no significance. Source data are provided as a Source Data file.

brush border of the linerixibat-treated mice (Fig. 4–k). Take together, these results suggest that ASBT is important for cholesterol absorption and NPC1L1 recycles in the ileum by mediating the entry of BAs into enterocytes.

## LIMA1 is required for NPC1L1 transportation but not ERC cholesterol egress

LIMA1 is a scaffold protein that bridges NPC1L1 to myosin Vb for NPC1L1 transportation to the PM[16,35]. BA decreased ERC cholesterol and caused the WT NPC1L1 transport to the PM (Fig. 5a). Although BA similarly redistributed cellular cholesterol, the NPC1L1-QKR mutant ($Q_{1277}$KR to AAA mutation), which failed to bind LIMA1[16], was still present in the ERC (Fig. 5a–c). Consistently, knockdown of *Lima1* entrapped NPC1L1 in the ERC but did not interfere with the effects of BAs on ERC cholesterol decrease (Fig. 5d–f), PM cholesterol increase or lipid droplet formation (Fig. 5g–j, Fig. S8). These results suggest that cholesterol decrease in ERC is upstream to NPC1L1 transportation.

## Depletion of ERC cholesterol recruits LIMA1 to ERC-localized NPC1L1

Next, we investigated the mechanism by which BAs triggered PM localization of NPC1L1. A short-term treatment of BA mixture, CDX or the four BA species (CDCA, DCA, GDCA or TDCA) could drastically transport cholesterol out of ERC and recruited intracellular LIMA1 to ERC, whereas NPC1L1 still stayed in ERC (Fig. 6a–c and S9a–c). As a control, TCA was unable to translocate LIMA1 to the NPC1L1-positive ERC (Fig. S9a–c). Since GDCA and TDCA were imported into cells by NTCP, they failed to induce LIMA1 relocation to ERC in the *Ntcp* silenced cells (Fig. 6d–f). CDCA or DCA entered cells in a manner independent of NTCP, and these two BAs could still cause ERC localization of LIMA1 in *Ntcp* knockdown cells (Fig. S9d–f).

The co-immunoprecipitation results revealed that GDCA, TDCA, BA mixture or CDX largely enhanced the interaction between NPC1L1 and LIMA1, but TCA failed to do so (Fig. 6g). Consistently, LIMA1 was not recruited to the ERC of the cells without NPC1L1 overexpression (Fig. S9g–i). The NPC1L1-QKR mutant did not recruit LIMA1 to ERC (Fig. S9j). The LIMA1 CLG mutant ($C_{164}$LG to AAA), which abolished NPC1L1-LIMA1 interaction[16], was unable to relocate to ERC (Fig. S9k).

## NPC1L1-QKR motif associates with ERC membrane in a cholesterol-dependent manner

The carboxyl terminus of NPC1L1 is a cytosolic flexible peptide containing $Q_{1277}$KR residues[16,17]. We hypothesized that the ERC cholesterol may regulate the interaction between membrane and NPC1L1 carboxyl terminus. We incubated the NPC1L1 carboxyl terminus (67 amino acids, named as NPC1L1-CT67) with liposomes containing different concentrations of cholesterol (Fig. 7a). The liposomes containing 30% cholesterol or higher pulled down NPC1L1-CT67, whereas the liposomes containing less cholesterol (no more than 20%) could not associate with the recombinant protein (Fig. 7a). We further identified

that the $A_{1272}$LAL residues, which are adjacent to $Q_{1277}$KR, were essential for the association with cholesterol-enriched liposomes (Fig. 7b and Fig. S10a–d).

In cultured cells, BA enhanced the interaction between NPC1L1 and LIMA1. However, the $A_{1272}$LAL deleted or mutated NPC1L1 variants showed constitutively strong binding to LIMA1 regardless of BA treatment (Fig. 7c). These NPC1L1 variants were prone to PM translocation (Fig.7d, e and S10e). Taken together, these results suggest that BAs carry away ERC cholesterol and cause $A_{1272}$LAL residues in NPC1L1 to dissociate from ERC membrane, which further allows $Q_{1277}$KR to bind to LIMA1 and then NPC1L1 is transported to PM.

## Discussion

The brush border membrane-localized NPC1L1 is required for cholesterol to enter enterocytes. Cholesterol is then transported to the ER where it is converted to CEs and packed into chylomicrons. Previous studies have found that NPC1L1 takes up intestinal cholesterol through vesicular endocytosis. The internalized cholesterol and NPC1L1 are present on Rab11-positive ERC that is spatially separated from the ER[11]. The decrease in ERC cholesterol liberates NPC1L1 that then recycles back to PM[10,35].

In this study, we find that BAs, specifically CDCA, DCA, GDCA, and TDCA, can mobilize cholesterol within the cell. The BA-mediated cholesterol diffusion is nondirectional as the cholesterol levels in both ER and PM are increased. ACAT2 is highly expressed in the ER of intestinal epithelial cells. So, ACAT2 converts cholesterol to CE and drives the ERC cholesterol mainly to ER. It has been known that intracellular cholesterol transport mechanisms are classed into vesicular and non-vesicular transport[36-38]. Non-vesicular cholesterol trafficking requires sterol transfer protein and often occurs at membrane contact sites between two adjacent membranes from different organelles, which is usually within 40 nanometers[39]. The BA-mediated cholesterol transport represents an uncharacterized cholesterol trafficking way, which may be specifically present in enterocytes or hepatocytes.

LDL-derived cholesterol (LDL-c) can be transported from lysosomes to PM/ER by multiple pathways[39,40]. The lysosomal proteins NPC1 and NPC2 are essential proteins for lysosomal cholesterol egress. However, mice lacking either NPC1 or NPC2 absorbed the same amount of cholesterol as WT mice[41]. Therefore, lysosomes seem unlikely play a role in dietary cholesterol transport. NPC1L1 inactivation by genomic knockout or ezetimibe decreased cholesterol absorption by ~70%[9]. Scavenger receptor class B type I (SR-BI), a glycoprotein that resides on apical and basolateral surfaces of the proximal intestinal villi, was suggested as a potential cholesterol transporter besides NPC1L1[42,43]. Mice overexpressing SR-BI in the intestine showed an increased cholesterol absorption[44]. The contributing ratio of SR-BI on dietary cholesterol absorption remains to be elucidated, since ablation of SR-BI in mice does not affect dietary cholesterol absorption and fecal cholesterol excretion[45]. We found that the loss-of-function

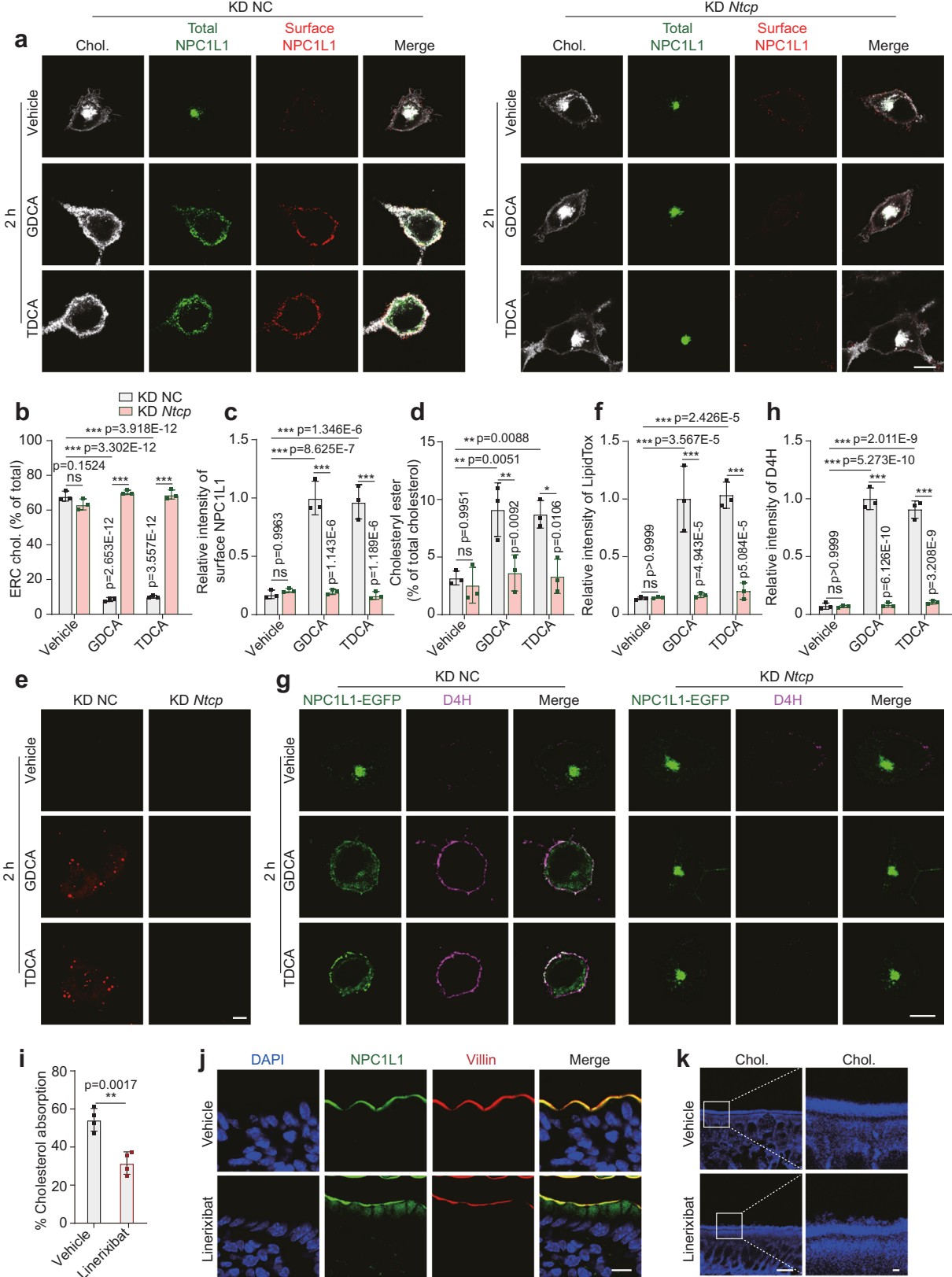

mutation of NPC1L1 endocytic sequence in mice reduced cholesterol absorption by ~53%, which is less than the ~70% decline of that in KO mice. Considering that the 4A-mutated NPC1L1 can still insert cholesterol into PM via its amino terminus (Fig. 1b), another cholesterol delivery route that conveys away cholesterol directly from PM may exist and compensate for cholesterol absorption. GRAMD/ASTER

facilitates cholesterol unidirectional transport from PM to ER by binding phosphatidylserine in the inner membrane of PM[46,47]. ASTERs are also enriched in the intestine and may be engaged in dietary cholesterol absorption.

BAs are biosurfactants that play key roles in lipid digestion and absorption in the gastrointestinal tract. On one hand, BAs absorb at

**Fig. 4 | NTCP is required for GDCA- or TDCA-mediated cholesterol transport in cells. a** Confocal images of CRL1601/NPC1L1-3×Myc-EGFP cells pretreated with vehicle, GDCA or TDCA. Scale bar, 10 μm. NC: negative control. Percentages of ERC cholesterol (**b**) and relative intensity of surface NPC1L1 (**c**) in (**a**) were quantified. The average intensity of surface NPC1L1 in GDCA-treated control cells was defined as 1. Values were presented as mean ± SD ($n = 3$ independent trials, 100 cells/trial). Two-way ANOVA with Tukey post hoc test, ***$P < 0.001$; ns, no significance. **d** Percentages of cholesteryl esters relative to total cholesterol in NC or *Ntcp* knockdown CRL1601/NPC1L1-3×Myc-EGFP cells with a 2-h incubation of indicated BAs. Values were presented as mean ± SD ($n = 3$). Two-way ANOVA with Tukey post hoc test, ***$P < 0.001$; ns, no significance. **e** LipidTox staining of CRL1601/NPC1L1-3×Myc-EGFP cells after 2-h incubation of vehicle, GDCA or TDCA. Scale bar, 10 μm. **f** Relative intensity of LipidTox in (**e**) was quantified. The average intensity of GDCA-treated control cells was defined as 1. Values were presented as mean ± SD ($n = 3$

independent trials, 100 cells/trial). Two-way ANOVA with Tukey post hoc test, ***$P < 0.001$; ns, no significance. **g** D4H staining of CRL1601/NPC1L1-3×Myc-EGFP cells with 2-h incubation of vehicle, GDCA or TDCA. Scale bar, 10 μm. **h** Relative intensity of D4H in (**g**) was quantified. The average intensity of D4H in GDCA-treated control cells was defined as 1. Values were presented as mean ± SD ($n = 3$ independent trials, 100 cells/trial). Two-way ANOVA with Tukey post hoc test, ***$P < 0.001$; ns, no significance. **i** Cholesterol absorption of the vehicle or linerixibat-treated mice was measured by dual isotope ratio method. Values were presented as mean ± SD ($n = 4$ mice). Unpaired two-tailed Student's *t* test. **$P < 0.01$. **j** Deparaffinized sections of the ileal intestine from the vehicle or linerixibat-treated nine-week-old NPC1L1-EGFP male mice ($n = 2$). Scale bar, 10 μm. **k** Filipin staining of ileal intestine sections from nine-week-old C57BL6J mice ($n = 2$). Scale bar, 10 μm (main); 1 μm (inset). Chol: Cholesterol. Source data are provided as a Source Data file.

lipid/water interfaces and promote the absorption of digestive enzymes onto fat droplets. On the other hand, BAs remove lipolytic products including fatty acids, monoacylglycerols and cholesterol from the interface by solubilizing them into mixed micelles. Previous studies have shown that TDCA displays a higher solubilizing capacity compared with TCA[48]. It is also reported that TCA shows a higher affinity for the lipid/water interface and may favor enzyme absorption, whereas TDCA displays a higher desorption ability from the interface and may effectively remove hydrolytic products from the interface[49]. These physicochemical characteristics are consistent with our data that TDCA (GDCA, DCA and CDCA) shows higher cholesterol mobilization ability than TCA in vitro and in cells (Figs. 2b, 3b).

The dietary cholesterol uptake efficiency in human populations varies from 29 to 80%[50,51]. This can be attributed to genetic variations of *NPC1L1*, *NUMB* or *LIMA1*[16,52,53]. Meanwhile, it is known that BAs are dynamically modulated, as BAs are conjugated with glycine, taurine and sulfate in the liver and primary BAs are metabolized to secondary BAs in the intestine[54]. The composition of BAs shows drastic individual difference. It has been reported that the percentage of GDCA and TDCA in bile can range from <5% to >30% or <1% to >10% among 20 healthy volunteers, respectively[55]. In current work, we find four BA species (CDCA, DCA, GDCA and TDCA), of a total of 12 main BA species in humans, can effectively transport cholesterol out of ERC (Fig. 1b–d and S2c–g). Thus, the diversity in BA composition may be another important reason determining the cholesterol absorption efficiencies among individuals.

We summarize the molecular pathway of intestinal cholesterol absorption (Fig. 7f). In the absence of cholesterol, NPC1L1 locates on the brush border membrane and the carboxyl terminal YVNxxF sequence associates with PM. With the help of BAs, cholesterol diffuses to brush border membrane in the form of cholesterol/BA micelles. The amino terminal domain of cholesterol binds cholesterol, induces the conformational change of NPC1L1 and causes the YVNxxF to dissociate from PM. Then NUMB binds to the motif and further recruits AP2/Clathrin[13]. Meanwhile, NPC1L1 constitutively interacts with Flotillins and locates in cholesterol-rich membrane microdomain. The internalization of NPC1L1-Flotillin carries a large amount of cholesterol into cells and transports to ERC[14]. BAs can be imported by passive diffusion in the duodenum and jejunum or by active transport involving ASBT in the ileum. The passive diffusion of BAs might play a more important role in cholesterol absorption, since a large amount of cholesterol absorption takes place in the upper intestine. BAs extract cholesterol from ERC and diffuse them throughout the cells. As the highly expressed ACAT2 converts cholesterol to CE, the ERC cholesterol is mainly transported to ER. In the ERC, the NPC1L1 carboxyl terminal fragment binds to cholesterol-rich membrane via $A_{1272}$LAL sequence. Cholesterol reduction in membrane causes $A_{1272}$LAL to separate from membrane, exposing the QKR motif to recruit LIMA1. LIMA1 bridges NPC1L1 to Myosin Vb and NPC1L1 is transported back to PM.

The apical membrane of enterocyte is capable of internalizing materials from intestinal lumen. Extensive apical endocytic membrane structures are observed in the neonatal rat and the pup absorbs dietary fat through endocytosis[56]. In postweaned animals, the small molecular fluorescent probe Alexa hydrazide is internalized and appears in sub-apical early endosomes with the similar morphology to the endocytosed NPC1L1[11,57]. Although it is still mysterious how fatty acids are absorbed in small intestine[58], fats can rapidly induce clathrin-dependent endocytosis, suggesting endocytosis might be a way for fatty acids to enter enterocytes[56,59]. For the cholesterol absorption, cholesterol is emulsified with BAs in intestinal lumen and diffuses to brush boarder membrane. Cholesterol is taken into ERC by NPC1L1-mediated endocytosis and BAs are transported into the cytosol by BA transporters on the apical membrane. Compared with pinocytosis in neonate, this mechanism ensures specificity. The intracellular BAs may facilitate transport of other hydrophobic molecules, such as fatty acids, between different organelles to promote absorption.

## Methods

### Animals

Male C57BL/6J mice (9 and 12 weeks of age) were purchased from Centers for Disease Control (Hubei, China). The *Npc1l1*-EGFP knock-in mouse, which containing an EGFP tag before the stop codon of *Npc1l1* in C57BL/6N mouse, was a kind gift from Prof. Weiping J. Zhang at Naval Medical University in China[60]. The Npc1l1-$Y_{1306}$VNxxF→AAAxxA knock-in mouse was generated from C57BL/6JGpt by CRISPR-Cas9 system (GemPharmatech). Neonatal NPC1L1-EGFP knock-in C57BL/6N male mice was used for in vitro intestinal culture assay and 12-week-old male knock-in mice were applied to bile duct cannulation assay. Littermates from the heterozygous of Npc1l1-$Y_{1306}$VNxxF→AAAxxA crossing were wild-type controls. Mice was housed in plastic cages in a specific pathogen-free animal facility under a 12-hour day/light cycle at 22 °C with a humidity of 50–60%. Mice were fed on chow diet (Beijing HFK Biosciences, 1026) *ad libitum*. No gender-based analyses have been performed in this study. We chose male mice in the study because they have been widely used to evaluate cholesterol absorption. In fact, cholesterol absorption efficiency is not significantly different between genders in mice[61]. Adult mice were sacrificed by cervical dislocation and neonatal mice were sacrificed by decapitation using a sharp pair of scissors.

### Ethics

All animal experiments were performed according to the protocols (WDSKY0201408) approved by the Institutional Animal Care and Use Committee of Wuhan University.

### Reagents

1,2-dioleoyl-snglycero-3-phosphocholine (DOPC) (850375), 1,2-dioleoyl-sn-glycero-3-phospho-L-serine (DOPS) (840035) and sphingo-myelin (LM2312) were from Avanti Polar Lipids. Ham's F-12 (Ham's F-12)

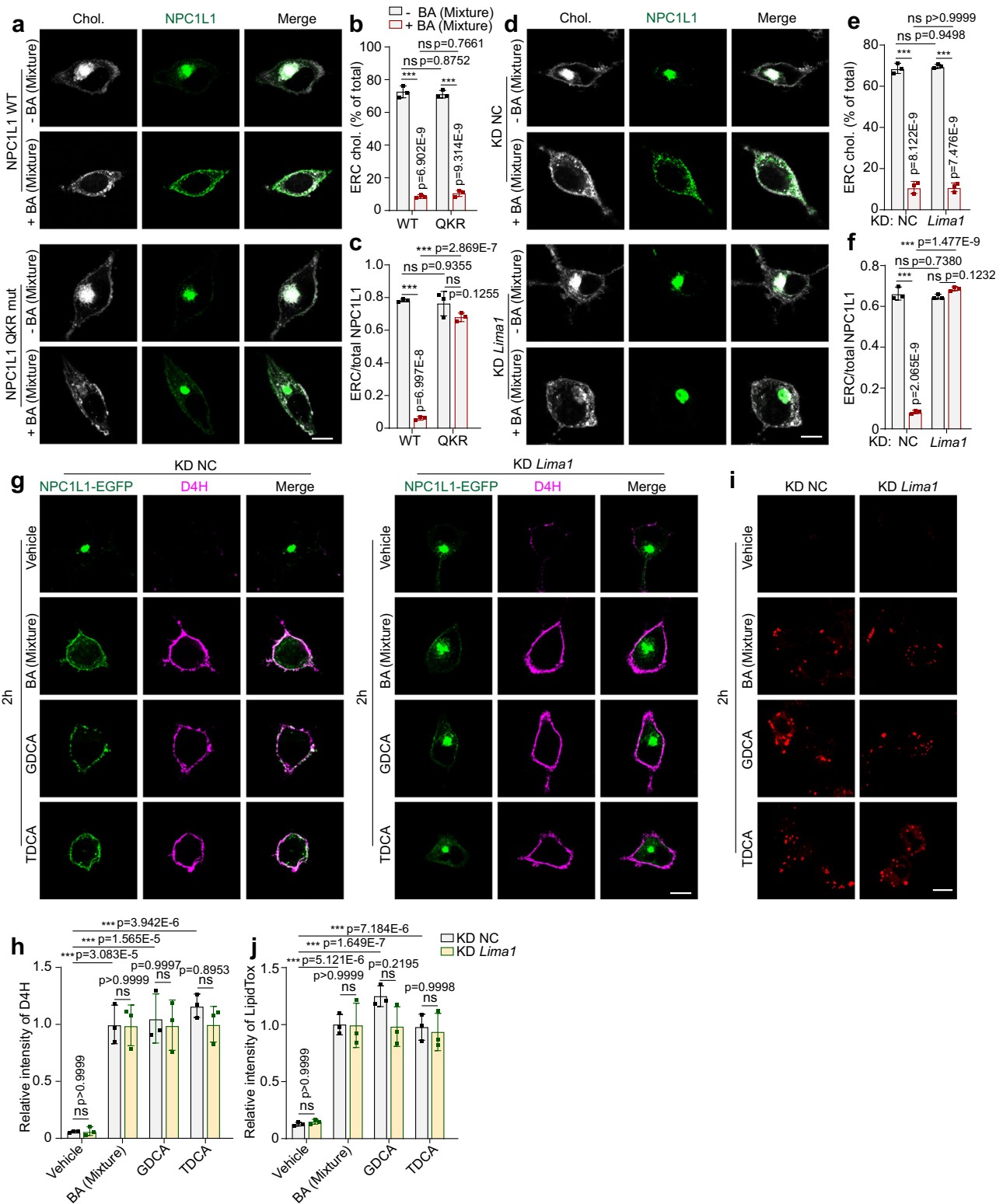

**Nature Communications** | (2023)14:6469

was from Gibco. Amplex Red Cholesterol Assay Kit (A12216) and HCS LipidTox neutral lipid stain (H34476) were from Invitrogen. Anti-FLAG M2 Affinity Gel (A2220), cholesterol (C8667), deoxycholate (D6750), filipin (F9765), glycocholate (700265 P), glycodeoxycholate (G9910), lithocholic acid (700218 P) and taurodeoxycholate (T0875) were from Sigma-Aldrich. The bile acid (BA) mixture (B822609) was from Macklin. Cholate (C426104), taurocholate (S100834) and taurolithocholate (S171146) were purchased from Alladin. Chenodeoxycholic Acid (HY-76847), gentamicin (HY-A0276A), glycochenodeoxycholate

(HY-N2334A), glycolithocholic acid (HY-116374), Linerixibat (HY-16643) and taurochenodeoxycholic acid (HY-N2027) were from Med Chem Express. PTFE Sub-Lite Wall Tube (SUBL-120, 0.012" OD x 0.006" ID), Teflon-coated guide wire (ST003, 0.003") and RenaSil Silicone Rubber Tubing (SIL-025, 0.025" OD x 0.012" ID) were purchased from Braintree Scientific Inc (MA). Cyclodextrin was purchased from Cyclodextrin Technologies Development. Total cholesterol assay kit and total triglyceride assay kit were purchased from Shanghai Kehua Bio-engineering Co., Ltd. Lipoprotein-deficient serum (LPDS, density

**Fig. 5 | NPC1L1-LIMA1 interaction is required for NPC1L1 transportation but not cholesterol egress. a** Confocal images of CRL1601 cells transfected with pCMV-NPC1L1-EGFP or $Q_{1277}KR \to AAA$ variant (QKR mut) with or without a 2-h treatment of BA mixture. Scale bar, 10 µm. Percentages of ERC cholesterol relative to the total cholesterol (**b**) and the relative intensity of NPC1L1 resided in ERC (**c**) in (**a**) were quantified. Values were presented as mean ± SD ($n = 3$ independent trials, 100 cells/trial). Two-way ANOVA with Tukey post hoc test, ***$P < 0.001$; ns, no significance. **d** Confocal images of control or *Lima1* knockdown CRL1601/NPC1L1-3×Myc-EGFP cells with or without a 2-h treatment of BA mixture. Scale bar, 10 µm. Percentages of ERC cholesterol relative to the total cholesterol (**e**) and the relative intensity of NPC1L1 resided in ERC (**f**) in (**d**) were quantified. Values were presented as mean ± SD ($n = 3$ independent trials, 100 cells/trial). Two-way ANOVA with Tukey post hoc test, ***$P < 0.001$; ns, no significance. **g** D4H staining prior to fixation was performed in control or *Lima1* knockdown CRL1601/NPC1L1-3×Myc-EGFP cells treated with vehicle, BA mixture, GDCA or TDCA for 2 h. Scale bar, 10 µm. **h** The relative intensity of D4H in (**g**) was quantified. The average intensity of D4H in BA mixture-treated control cells was defined as 1. Values were presented as mean ± SD ($n = 3$ independent trials, 100 cells/trial). Two-way ANOVA with Tukey post hoc test, ***$P < 0.001$; ns, no significance. **i** LipidTox staining of control or *Lima1* knockdown CRL1601/NPC1L1-3×Myc-EGFP cells after 2-h incubation of vehicle, BA mixture, GDCA or TDCA. Scale bar, 10 µm. **j** The relative intensity of LipidTox in (**i**) was quantified. The average intensity of BA mixture-treated control cells was defined as 1. Values were presented as mean ± SD ($n = 3$ independent trials, 100 cells/trial). Two-way ANOVA with Tukey post hoc test, ***$P < 0.001$; ns, no significance. Chol: Cholesterol. Source data are provided as a Source Data file.

>1.215 g/mL) was prepared from newborn calf serum by ultra-centrifugation in our laboratory.

## Plasmids

The coding sequence of human LIMA1 was inserted into p3×FLAG-CMV14 or pmCherry-N1 to generate pCMV14-LIMA1-3×FLAG (encoding LIMA1-FLAG) or pCMV-LIMA1-mCherry (encoding LIMA1-mCherry), respectively. To generate pCMV-NPC1L1-3×Myc-EGFP plasmid, the coding sequence of human NPC1L1 with an insertion of 3×Myc tag after $966^{th}$ amino acid was fused with a C-terminal EGFP and then cloned into pcDNA3. The deletion ($\Delta A_{1272}LAL$) or mutation ($A_{1272}LAL$-to-$E_{1272}EEE$ substitution, $A_{1272}LAL \to 4E$) of pCMV-NPC1L1-3×Myc-EGFP were generated by overlap extension PCR. The coding sequence of NPC1L1-3×Myc-EGFP-$\Delta A_{1272}LAL$ or NPC1L1-3×Myc-EGFP-$A_{1272}LAL \to 4E$ was also inserted into pLVX-IRES-puro vector by standard PCR, respectively. The sequence of mouse Rab11a was cloned into pEGFP-C1 or pmCherry-C1 to encode EGFP-Rab11a or mCherry-Rab11a, respectively. The sequence of human TFR was cloned into pmCherry-N1 to generate pCMV-TFR-mCherry. The coding region of human NPC1L1-CT67 (1272-1275 a.a.) was amplified from pCMV-NPC1L1-3×Myc-EGFP by standard PCR, and then inserted into the pET28a to obtain pT7-His$_6$-NPC1L1-CT67. The truncations of pT7-His$_6$-NPC1L1-CT67 ($\Delta$1-10, $\Delta$11-20, $\Delta$21-30, $\Delta$31-40, $\Delta$41-50, $\Delta$51-60, $\Delta$21-30, $\Delta$61-67, $\Delta$GPD, $\Delta$VNP and $\Delta$ALAL) were prepared by over extension PCR. The D4H, domain 4 of Perfringolysin O carrying D434S mutation, fused with N-terminal His$_6$-mCherry was inserted into pET28a vector[62].

## Antibodies

The rabbit polyclonal antibody against NPC1L1 (1 µg/mL for immunoblotting and 5 µg/mL for immunohistochemistry) was generated in our laboratory[11]. The mouse monoclonal antibody against SREBP2 (5 µg/mL for immunoblotting) was produced and purified from hybridoma cell line 1D2 (ATCC, Cat Num #CRL-2545) in our laboratory. The information about commercial antibodies is listed in Supplementary Table 2.

## Cell culture

CRL1601 cells were obtained from ATCC (Cat# CRL1601). CRL1601, CRL1601/NPC1L1-3×Myc-EGFP (CRL1601 cells stably expressing NPC1L1-3×Myc-EGFP), CRL1601/NPC1L1-3×Myc-EGFP-$\Delta A_{1272}LAL$ (CRL1601 cells stably expressing NPC1L1-3×Myc-EGFP with a deletion of $\Delta A_{1272}LAL$) and CRL1601/NPC1L1-3×Myc-EGFP-$A_{1272}LAL \to 4E$ (CRL1601 cells stably expressing NPC1L1-3×Myc-EGFP with $A_{1272}LAL$-to-$E_{1272}EEE$ substitution) were grown in a monolayer at 37 °C with 5% CO$_2$. Cells were maintained in the Dulbecco's Modified Eagle Medium (DMEM) containing 100 units/mL penicillin and 100 µg/mL strepto-mycin sulfate and 10% fetal bovine serum (FBS). To generate CRL1601/NPC1L1-3×Myc-EGFP-$\Delta A_{1272}LAL$ or CRL1601/NPC1L1-3×Myc-EGFP-$A_{1272}LAL \to 4E$ cell, CRL1601 cells were transfected with lentivirus expressing NPC1L1-3×Myc-EGFP-$\Delta A_{1272}LAL$ or CRL1601/NPC1L1-3×Myc-EGFP-$A_{1272}LAL \to 4E$, respectively.

The cholesterol starvation medium was composed of DMEM supplemented with 1 µmol/L lovastatin, 10 µmol/L mevalonate and 5% (w/v) LPDS. For BA treatment assay, cells were washed with 1×PBS once and then incubated with DMEM supplemented with 5% LPDS in the presence or absence of indicated BAs for 10 min or 2 h. For transient overexpression assay, CRL1601 and CRL1601/NPC1L1-3×Myc-EGFP cells were transfected with indicated plasmids using FuGENE HD Transfection Reagent (Promega, E2311) according to the technical manual. Briefly, 1 µg recombinant plasmids and 3 µL FuGENE HD Transfection Reagent were mixed together for 15 min in 100 µL opti-MEM I Reduced Serum Medium (Gibco, 11058021). Six hours later, the media was replaced. After 72 h, cells with indicated treatment were harvested for WB (collected by Tanon Imager 5200 software v2.03) or immunostaining.

## RNA interference

On day 0, CRL1601/NPC1L1-3×Myc-EGFP cells were seeded on 12-well plates or 60 mm dishes at a confluency of 30%. Twenty-four hours later, the cells were transfected with negative control siRNA, siRNA against rat *Ntcp1* or *Lima1* for 48 h using Lipofectamine RNAiMAX Transfection Reagent (Invitrogen, 13778150). Briefly, siRNA was dissolved into 10 µmol/L with RNase-free water. 2 µL siRNA and 3 µL transfection reagent were separately added into 75 µL opti-MEM I Reduced Serum Medium for 5 min, then the diluted reagent and siRNA were mixed together and incubated for 5 min. Replace the culture medium with the opti-MEM I Reduced Serum Medium and add siRNA-lipid complex into the medium. Before harvesting, cells were treated with DMEM containing 5% LPDS plus indicated BAs for 10 min or 2 h as described in the Figure legends. The qPCR was performed in Bio-Rad CFX Connect Real-Time System and analyzed by CFX Manager v3.1. The sequence of siRNAs and qPCR primers used in this study are listed in Supplementary Table 1.

## Bile acid treatment

The BA mixture from *Sus Scrofa* was dissolved with 50% ethanol to prepare the 25 mg/mL BA mixture stock solution. The BA mixture stock solution was filtrated with a 0.22 µm filter and diluted with 5% LPDS in DMEM at the ratio of 1:100 to yield a working medium with 0.25 mg/mL BA mixture.

CA, CDCA, DCA and LCA stock solutions were dissolved in DMSO at a concentration of 100 mmol/L. GCA, TCA, GDCA, TDCA, GCDCA, TCDCA, GLCA and TLCA were dissolved in water at a concentration of 100 mmol/L. When in use, the stock solutions of each BA species were diluted 1000 times in DMEM with 5% LPDS and the final working concentration of these bile acids is 0.1 mmol/L. 1.5% or 3% w/v CDX in DMEM harboring 5% LPDS was used to treat cells for 2 h or 10 min as indicated in the Figure legends.

To test the cholesterol mobilizing ability of bile acids, CRL1601/NPC1L1-3×Myc-EGFP cells were treated with DMEM containing 5% LPDS (designed as vehicle), 250 µg/mL BA, or 0.1 mmol/L indicated bile acids in the vehicle medium for 2 h prior to harvest,

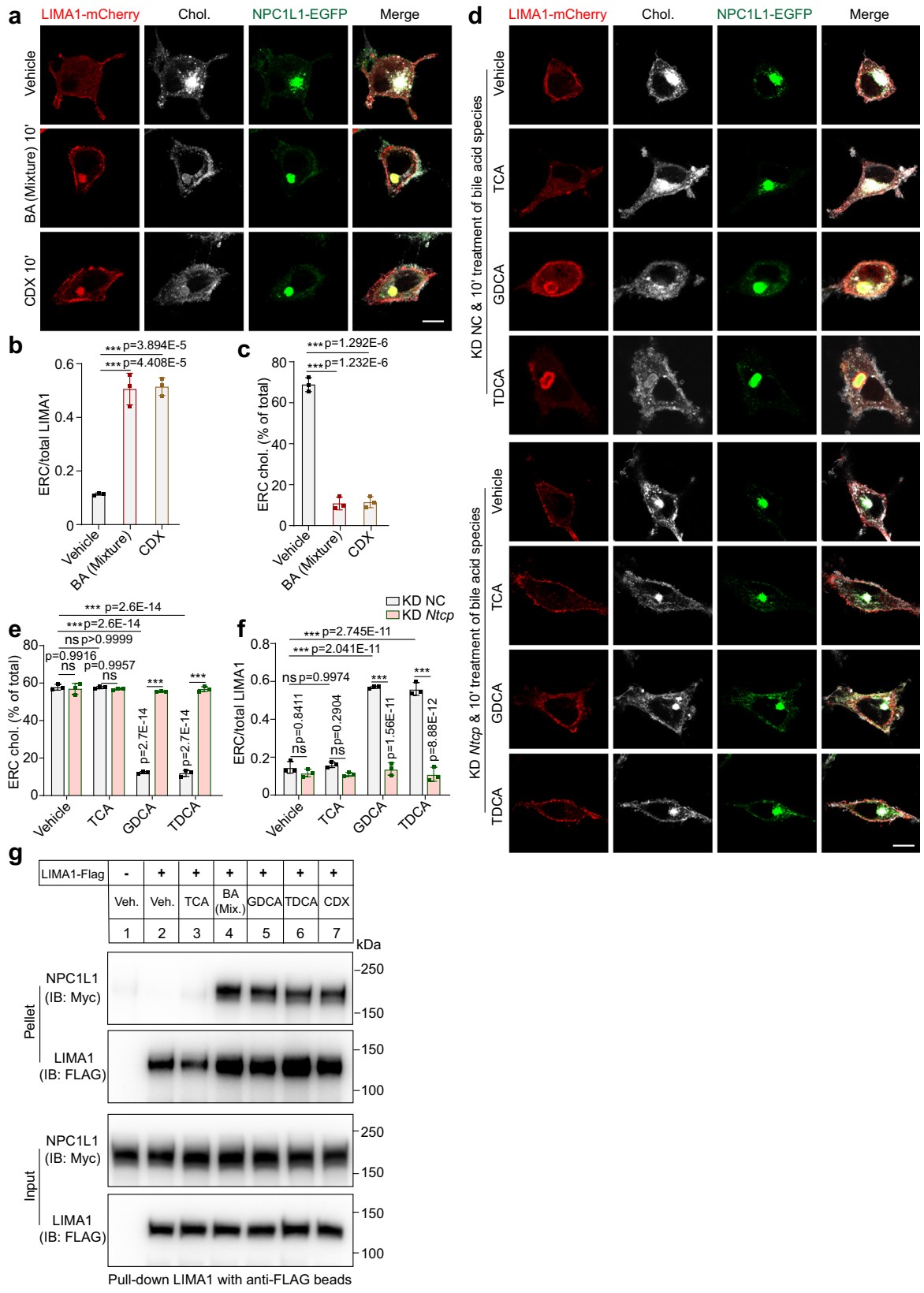

respectively. If needed, the avasimibe group was added additional 10 µmol/mL avasimibe during the 2-hour incubation. In the short-term treatment of bile acids, CRL1601 or CRL1601/NPC1L1-3×Myc-EGFP cells were treated with control medium, BA, indicated bile acids or 3% w/v CDX for 10 min prior to 4% paraformaldehyde (PFA) fixation, respectively.

## LipidTox staining

For LipidTox staining of CRL1601/NPC1L1-3×Myc-EGFP cells, cells grown on coverslip were washed with 1×PBS once, and fixed with 4% paraformaldehyde for 30 min. Cells were washed with 1×PBS for 3 times and incubate with 1×PBS containing 0.1% v/v HCS LipidTox neutral lipid stain for 30 min. Remove the buffer from the cells and

**Fig. 6 | Acute decrease of cholesterol in ERC relocates LIMA1 to NPC1L1-positive ERC. a** A 10-min treatment of vehicle, BA mixture or CDX in CRL1601/NPC1L1-3×Myc-EGFP cells transfected with pCMV-LIMA1-mCherry. Scale bar, 10 µm. The relative intensity of LIMA1-mCherry (**b**) and cholesterol (**c**) resided in ERC in (**a**) was quantified. Values were presented as mean ± SD (*n* = 3 independent trials, 100 cells/trial). Two-way ANOVA with Tukey post hoc test, ***P < 0.001. ns, no significance. **d** Control or *Ntcp* knockdown CRL1601/NPC1L1-3×Myc-EGFP cells with a 10-min incubation of vehicle, indicated BAs. Scale bar, 10 µm. The relative intensity of cholesterol (**e**) and LIMA1-mCherry (**f**) resided in ERC in (**d**) was quantified. Values were presented as mean ± SD (*n* = 3 independent trials, 100 cells/trial). Two-way ANOVA with Tukey post hoc test, ***P < 0.001. ns, no significance. **g** CRL1601/NPC1L1-3×Myc-EGFP cells were transfected with pCMV-LIMA1-FLAG for 48 h and subjected to the co-immunoprecipitation by the anti-FLAG agarose after a 10-min treatment of indicated BAs or 3% CDX (*n* = 2). Chol., Cholesterol; Veh, Vehicle; Mix., Mixture. Source data are provided as a Source Data file.

wash cells with 1×PBS for 3 times. The coverslips containing cells were mounted with 50% (v/v) glycerol and the edges of coverslips were sealed with nail polish.

For LipidTox staining of intestinal slices, 10-µm-thick frozen sections in the glass slides were incubated at 37 °C for 30 min and washed with 1×PBS for 3 times. The sections were permeabilized with 0.2% v/v triton in 1×PBS for 10 min, washed with 1×PBS for 3 times, and incubated with primary antibody diluted in 1×PBS plus with 1% w/v BSA at 4 °C overnight. After 3 times washing of 1×PBS, the sections were incubated with the secondary antibody diluted in 1% (w/v) BSA at room temperature (RT) for 1 h. The sections were washed with 1×PBS 3 times, stained with 2% v/v HCS LipidTOX neutral lipid stain in 1×PBS at RT for 50 min, and washed with 1×PBS for 5 times. A clean coverslip was mounted onto the sections with 50% (v/v) glycerol and the edges of the coverslip were sealed with nail polish.

### Liposome preparation and co-sedimentation assay

100-nm liposomes composed of DOPC, DOPS and sphingomyelin with or w/o cholesterol of different concentrations were prepared as previously described[63]. Briefly, a total of 1 mg lipid mixture composed of DOPC, DOPS and sphingomyelin with or w/o cholesterol in the indicated molar ratio as described was resuspended in 1 mL chloroform and pipetted into a 50-mL round bottom flask. The organic solvent in the flask was dried at 37 °C in a rotary evaporator with a rotating speed of 50 r/min. Lipids were resuspended in 1 mL 20 mmol/L Hepes (pH 7.2) and then subjected to 5 cycles of fast freeze-thaw test by placing sample vial in liquid nitrogen for 1 min and a 37 °C water bath for 2 min. To generate liposomes, the lipid suspension was forced through a 0.1 µm polycarbonate filter for 21 times by the mini-extruder (Avanti Polar Lipids, 610000-1EA).

To identify the interactions between the liposomes and WT or mutant forms of $His_6$-NPC1L1-CT67 protein, 40 µL liposomes and 10 µL proteins (0.75 µg/µL stock solution) were mixed in 200 µL HEK buffer (20 mmol/L HEPES, pH 7.2, 120 mmol/L potassium acetate and 1 mmol/L $MgCl_2$) and then incubated for 30 min at 37 °C. Liposomes were spun down at 250,000 × g for 30 min. Supernatants (designed as liposome unbound proteins) and pellets (designed as liposome bound proteins) were harvested and applied to western blotting analysis.

To assess the diameter of liposomes, 25 µg liposomes together with indicated bile acids were incubated in 1×PBS with a total volume of 100 µL at RT for 2 h. Then the liposomes were diluted in 2 mL 1×PBS and applied to diameter determination by Zetasizer Nano ZSP (Malvern Instruments, Malvern UK) using a detection angle of 173° at a temperature of 25 °C. The Nano ZSP uses a 10 mW He-Ne laser operating at a wavelength of 633 nm.

### In vitro cholesterol mobilizing assay

Liposomes containing DOPC, DOPS and sphingomyelin in the presence or absence of cholesterol in a molar ratio of 2:5:3 or 2:1:3:4 were applied to the mobilizing assay. Liposomes with or w/o cholesterol were separated by a semi-permeable membrane with the pore size allowing the diffusion of molecules below 50 kDa. 0.25 mg/mL bile acids mixture or 0.1 mmol/L indicated bile acid species were added into the cholesterol-liposome side and the permeator was incubated for 3 h at 4 °C on a shaker. Samples from the other side were collected

and cholesterol content was measured using Amplex Red Cholesterol Assay Kit following the manufacturer's instructions. The relative amount of mobilized cholesterol is the specific value that the total cholesterol mobilized to the non-cholesterol liposome divided by the total cholesterol in the cholesterol-liposome.

### Measurement of total cholesterol and cholesteryl esters

Cells were homogenized in chloroform/methanol (2:1) to extract lipids and then centrifuged at 20,000 × g for 10 min. The organic phase was harvested and dried using nitrogen flow. The pellets were resuspended and applied to cholesterol and cholesteryl esters with Amplex Red Cholesterol Assay Kit. To measure cholesteryl esters, parallel reactions were made with the presence or absence of cholesterol esterase. The readings were recorded as total cholesterol and free cholesterol respectively. And the difference was calculated as cholesteryl esters.

### Bile duct cannulation

12-week-old male C57BL/6J mice or NPC1L1-EGFP knock-in mice were anesthetized with tribromoethanol (0.02 mL/g of 1.2% stock solution) by intraperitoneal injection. Then a small incision was made in the abdominal wall near the duodenal site to visualize the common bile duct. A small hole in the common bile duct was made with the tip of a 29 G insulin syringe. Then PTFE Sub-Lite Wall Tube pre-loaded with a Teflon-coated guide wire was inserted into the common bile duct. After successful catheterization, the cannula was secured with surgical sutures. A guide wire was slowly removed from PTFE tubing and bile would slowly flow into the cannula. Then a softer and larger RenaSil Silicone Rubber Tubing was attached to the PTFE tubing and secured with sutures. The tube was tunneled subcutaneously through the abdominal wall and exteriorized in the scapular region. Sutures were applied to both wound sites to secure the catheter in place. The bile was collected into a tube placed on its back. Mice were placed on a 37 °C heating pad until woke up. Mice were singly housed after surgery.

### Intestinal filipin staining

Intestinal samples from *Npc1l1-EGFP* knock-in mice were washed with ice cold 1×PBS and fixed in 4% PFA at 4 °C for 12 h. The sample was washed with 1×PBS for 3 times and soaked in 30% sucrose for 48 h prior to embedding and freezing in Tissue-Tek OCT (Sakura, 4583). Then the intestine was cut into 10-µm thin sections, and these sections were placed on a hot plate at 37 °C for 1 h. Finally, the sections were stained with 50 µg/mL filipin diluted in 1×PBS for 40 min at RT and mounted in coverslips after 5 times washing in 1×PBS.

### In vitro intestinal culture

The in vitro intestinal culture was conducted as described[20]. The small intestine of the neonatal mouse was longitudinally opened and gently rinsed with ice-cold 1×PBS. The intestine was cut into about 1 cm segments, then intestinal segments were embedded in 1 mL type I collagen gel solution (Wako, 637-00653) and placed into a 30-mm cell culture insert (Millicell, PIHP03050), which had been pre-covered with 1.5 mL type I collagen gel solution. The cell culture insert was incubated with 2 mL Ham's F12 medium containing 20% (v/v) FCS and 50 µg/mL gentamicin in a 60 mm dish, and the

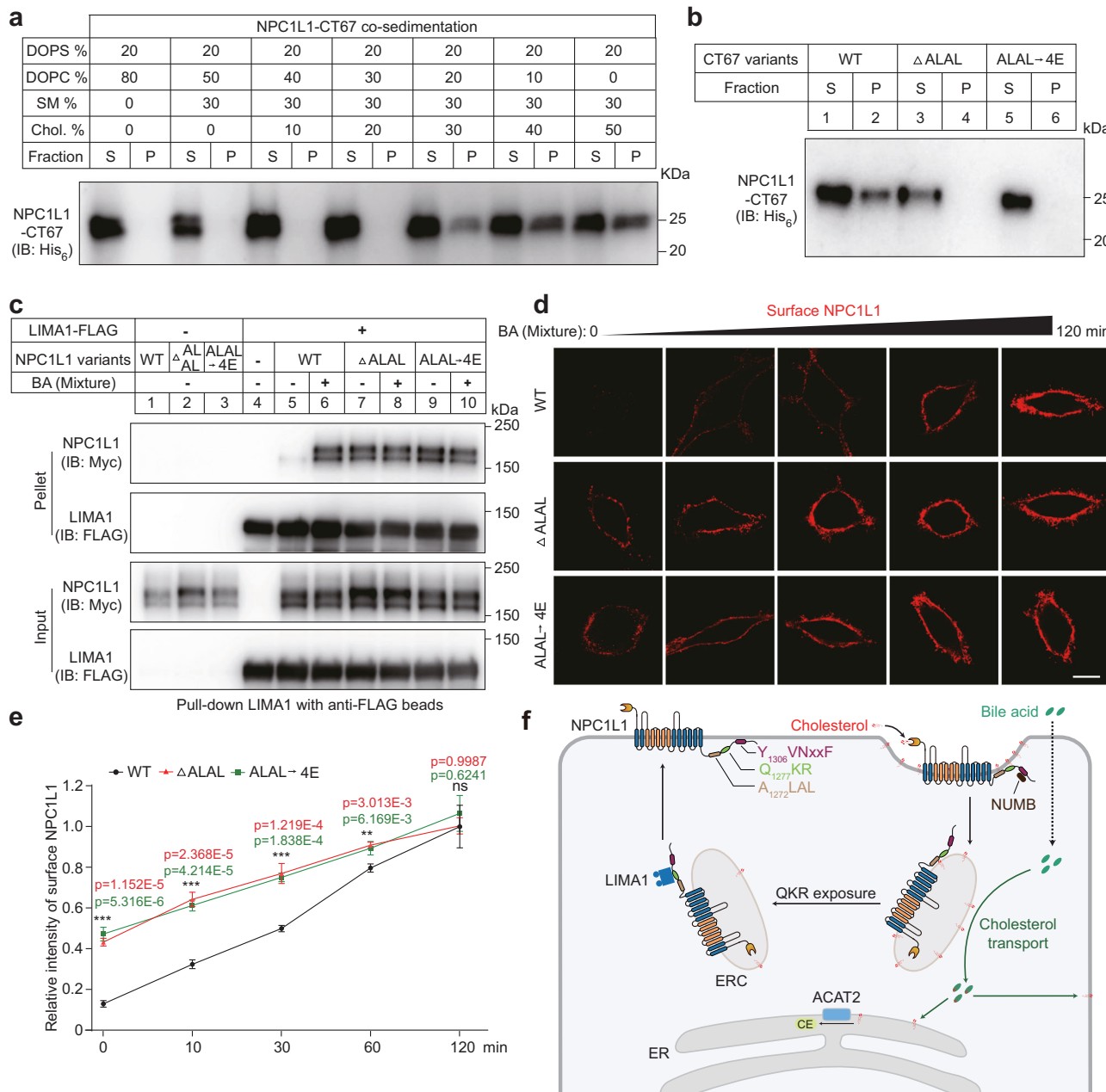

**Fig. 7 | NPC1L1 associates with membrane in a cholesterol-dependent manner.**
**a** NPC1L1-CT67 recombinant protein was purified and co-sedimented with serial liposomes with different concentrations of cholesterol ($n = 2$). After incubation and centrifugation, supernatant (S) and pellet (P) were collected and subjected to immunoblotting analysis. **b** The purified NPC1L1-CT67 recombinant proteins (WT, $\Delta A_{1272}LAL$ and $A_{1272}LAL \rightarrow 4E$ mutant) were analyzed using liposome co-sedimentation assay ($n = 2$). **c** Interaction of LIMA1-FLAG and indicated NPC1L1 variants (WT, $\Delta A_{1272}LAL$ and $A_{1272}LAL \rightarrow 4E$) was detected in the presence or absence of BA mixture ($n = 2$). **d** Representative confocal images of surface NPC1L1 of CRL1601 cells overexpressing NPC1L1-3×Myc-EGFP or $A_{1272}LAL$ deletion ($\Delta A_{1272}LAL$) or mutation ($A_{1272}LAL \rightarrow 4E$) variants following BA mixture treatment over time. Scale bar, 10 μm. **e** The relative intensity of surface NPC1L1 in (**d**) was analyzed. The average intensity of cells overexpressing WT NPC1L1-3×Myc-EGFP with 120 min-treatment of BA mixture was defined as 1. Values were presented as mean ± SD ($n = 3$ independent trials, 100 cells/trial). One-way ANOVA with Tukey post hoc test, ***$P < 0.001$. **f** A working model illustrating the role of BAs involved in NPC1L1-mediated cholesterol transport. CE cholesteryl ester, ER endoplasmic reticulum, ERC endocytic recycling compartment. Source data are provided as a Source Data file.

intestinal segments were grown at 37 °C with 5% $CO_2$. On day 4, the 60-mm dishes were washed with 1×PBS and added with 2 mL Ham's F12 medium containing 10% (v/v) LPDS plus 10 μg/mL cholesterol in the presence or absence of indicated bile acids or BA mixture. If needed, 20 μmol/L avasimibe was added to the LPDS medium. After 2 h incubation, the intestinal segments were carefully collected, fixed with 4% PFA at 4 °C overnight, and then embedded in Tissue-Tek OCT.

## D4H staining

CRL1601/NPC1L1-3×Myc-EGFP cells on the coverslips were washed with ice-cold 1×PBS for once and then incubated with 10 μg/mL recombinant mCherry-D4H (dissolved with 1% w/v BSA) for 30 min on ice. The samples were rinsed with ice-cold 1×PBS for 3 times and fixed with 4% PFA at RT. Coverslips were mounted on glass slides using FluorSave (Millipore, 345789) and imaged on a Leica Biosystems SP8 laser scanning microscope.

## Immunofluorescence

CRL1601 or CRL1601/NPC1L1-3×Myc-EGFP cells were grown on glass coverslips and fixed with 4% paraformaldehyde (PFA) for 30 min. For impermeable NPC1L1-3×Myc-EGFP staining, cells were serially incubated with primary antibody against Myc tag (1:500 dilution, 1% w/v BSA) and appropriate secondary antibodies (1:500 dilution, 1% w/v BSA) at RT for 1 h, respectively. Then cells were immuno-stained with filipin at RT for another 1 h. The coverslips were washed with 1×PBS for 3 times and mounted by FluorSave.

## Quantification and analysis of immunofluorescent images

All confocal images were acquired by a Leica Biosystems SP8 laser scanning microscope with Leica LAS X (v3.5.2.18963). A total of 300 cells from 3 independent trials were quantified, and the average of 100 cells from the same trial represented one statistic on bar chart. For quantification of the relative fluorescent intensity of LipidTox, D4H and surface NPC1L1, the contours of cells were defined manually, and then the background-subtracted fluorescence intensity of LipidTox, D4H or surface NPC1L1 within cell was quantified using Image J (v1.50i). The relative intensity of D4H or surface NPC1L1 was normalized to BA, CDCA or GDCA treated cells as indicated in the relevant legends.

The ERC cholesterol was defined as the cholesterol colocalized with intracellular NPC1L1. To quantify the ERC cholesterol, the cholesterol confocal images were converted to grayscale and subtracted background by the threshold tool. Then, ERC was manually outlined according to their contours in corresponding images. The contour line, namely region of interest (ROI), was copied to the background-subtracted cholesterol image by the ROI manager tool, and the intensity of ROI in the cholesterol images was determined to obtain the intensity of ERC cholesterol. Finally, we measured the cholesterol intensity of the whole cell and calculated the percentage of ERC cholesterol by the ratio.

Similarly, the ERC NPC1L1 referred to intracellular NPC1L1 residing on ERC, and the total NPC1L1 means the whole cell NPC1L1. The contour of ERC or the cell was copied from the image to generate the ROI of ERC NPC1L1 or total NPC1L1, respectively. The intensity of the two ROIs was measured and applied to ERC/total NPC1L1 calculation.

## Flow cytometry

CRL1601/NPC1L1-3×Myc-EGFP cells were washed with 1×PBS 2 times, suspended using 2 mM EDTA in 1×PBS and then rinsed with ice-cold 1×PBS once. CRL1601/NPC1L1-3×Myc-EGFP cells were fixed with 4% PFA for 30 min at 4 °C. Next, cells were incubated with anti-Myc antibody and secondary antibody (diluted with 1% w/v BSA) at 4 °C for 1 h in turn. Samples were washed with 1×PBS 3 times, resuspended with 1% BSA in 1×PBS, and then detected by CytoFLEX (Beckman Coulter) with CytExpert (v2.4.0.28). The data from flow cytometry were analyzed by FlowJo (v10.0.7). The gating strategy for analysis of surface NPC1L1 is shown in Figure S11. A total of 6,000 cells from 3 independent trials were counted.

## Quantitative real-time PCR

The total RNA was extracted from CRL1601 cells transfected with indicated siRNAs by Trizol as described[64]. 2 μg mRNA from CRL1601 cells was used for cDNA synthesis and the relative mRNA level of indicated genes was analyzed by the comparative CT method. Rat *Gapdh* was used as the control.

## Co-Immunoprecipitation assay

CRL1601 cells or CRL1601/NPC1L1-3×Myc-EGFP cells were transfected with LIMA1-FLAG and WT or mutant forms of NPC1L1-3×Myc-EGFP as indicated in legends. Cells were treated with BA mixture or indicated bile acid species for 10 min prior to harvest. Then cells were homogenized in NP40 buffer (0.5% NP40 in PBS containing 5 mmol/L EDTA and EGTA) and centrifuged at 3,000 × g for 10 min. The protein concentrations in the lysates were determined by Pierce™ BCA Protein Assay Kits (Thermo Scientific, Cat# 23227) and analyzed by Bio-Rad iMark Microplate Absorbance Reader with Microplate Manager 6 Software (v6.3). The lysates with equivalent protein amounts were then precleared with protein A/G beads at 4 °C for 30 min and centrifuged at 1,000 g for 3 min. The supernatant was incubated with anti-FLAG beads at 4 °C for 2 h. The anti-FLAG beads were spun down at 1000 × g for 3 min and washed with NP40 buffer 5 times. Finally, the beads were mixed with SDS-PAGE Protein Loading Buffer and examined by immunoblotting analysis.

## Immunohistochemistry analysis

Intestinal samples from euthanized male mice were fixed with 4% PFA, dehydrated through a series ethanol of different concentration gradients, cleared in xylene, and embedded in paraffin. Samples were cut into 10-μm sections and deparaffinized, then samples were heated at 95 °C for 15 min with Tris-EDTA buffer (20 mmol/L tris base, 1 mmol/L EDTA, pH 9.0) to retrieve antigens. Samples were permeabilized and blocked in PBS with 5% FBS and 0.5% Triton X-100 for 1 h at RT. Sections were then incubated with primary antibodies overnight at 4 °C. After washing with PBS for 5 times, sections were incubated with appropriate Alexa Fluor dye-conjugated secondary antibodies for 1 h at RT. Sections were finally counterstained with Hoechst (Invitrogen) and mounted using FluorSave (Millipore). Images were acquired on a Leica TCS SP8 confocal microscope.

## Cholesterol absorption measurement

The cholesterol absorption assays were conducted as described[16]. The nine-week-old male *Npc1l1*-$Y_{1306}$VNxxF→AAAxxA mice were orally gavaged with 100 μL corn oil containing 5 μCi [$^3$H] cholesterol and 0.1 mg unlabeled cholesterol. After 2 h, cholesterol was extracted from the plasma and livers of the mice and counted using the liquid scintillation counter. The percentage cholesterol absorption was calculated as follows: % cholesterol absorption = [$^3$H] in liver (or plasma)/ [$^3$H] dosing × 100. Total radioactivity in the plasma was calculated based on the assumption that mice possess approximately 4 mL plasma per 100 g body weight.

For the fecal dual isotope ratio assay, nine-week-old male C57BL/6J mice were pretreated with vehicle or 0.2 mg/mL linerixibat once daily for 3 days prior to cholesterol gavage, and continued to the end of the fecal collection. On the third day, mice were orally gavaged with 100 μL corn oil containing [$^{14}$C]-cholesterol (0.5 μCi), [$^3$H]-sitosterol (1 μCi) and 0.1 mg unlabeled cholesterol at 2 h after linerixibat treatment. Then the feces from the mice within 2 days were gathered. The [$^{14}$C]-cholesterol and [$^3$H]-sitosterol in the feces were extracted in chloroform and methanol mixture (with a ratio of 2 to 1) and determined by liquid scintillation counter. The percent cholesterol absorption was calculated as follows: % cholesterol absorption = ([$^{14}$C]/ [$^3$H] dosing mixture – [$^{14}$C]/[$^3$H] feces)/([$^{14}$C/[$^3$H] dosing mixture) × 100.

## Statistical analysis

Data were analyzed by GraphPad Prism 9.0 and expressed as mean ± SD. Statistical comparisons between two groups were conducted by Unpaired two-tailed Student's *t* test. Comparisons for more than two groups were conducted using a One-way analysis of variance (ANOVA) with Tukey's post hoc test. Comparisons for two categorical independent variables that each have multiple levels were performed using Two-way ANOVA with Tukey's post hoc test. Sample sizes, statistical tests and exact *P* values (calculated to fifteen decimal places) for each experiment are described in the relevant figure or Source Data file. The *P* values exceeding 0.05 were considered statistically not significant.

## Reporting summary

Further information on research design is available in the Nature Portfolio Reporting Summary linked to this article.

## Data availability

A Reporting Summary for this article describing the experiment details is available as Supplementary Information file. The data supporting the findings of this study are available within the paper and its supplementary information. Source data underlying figures and supplementary figures are provided as a Source Data file. Specific data *P* values are also included within the Source Data file. Source data are provided with this paper.

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

## Acknowledgements
We thank Ms. Dan Liang and Jing Jin for technical assistance. This work was supported by grants from National Natural Science Foundation of China (91954203 and 32021003), 111 Project of the Ministry of Education of China (B16036), the China Postdoctoral Science Foundation Grant (2021M692478) and the Fundamental Research Founds for the Central Universities (2042022kf1045). B.-L. Song acknowledges the support from the Tencent Foundation through the XPLORER PRIZE.

## Author contributions
B.-L.S. conceived and directed the project. J.X., L.-W.D. and B.-L.S. designed the experiments. J.X., L.-W.D., S.L., F.-H.M., C.X. and X.-Y.L. performed the experiments. W.Z. provided the NPC1L1-EGFP knock-in mice. J.X., L.-W.D., J.L. and B.-L.S. analyzed the data, wrote the paper with input from all authors.

## Competing interests
The authors declare no competing interests.
