## [Peer Review File · Nature Communications]

Bile acids-mediated intracellular cholesterol transport promotes intestinal cholesterol absorption and NPC1L1 recyclingREVIEWER COMMENTS

Reviewer #1 (Remarks to the Author):

This interesting paper addresses (patho)physiologically important questions regarding the mechanisms that drive transport of (relatively large amounts of) absorbed cholesterol from ERC to ER inside enterocytes, thereby allowing the cholesterol to be esterified by ACAT2 for incorporation into chylomicrons or lipid droplets and, at the same time, relocation of NPC1L1 to the plasma membrane for uptake of additional cholesterol. Authors propose a novel “intracellular cholesterol transport function of bile acids” implying that specific bile acids, i.e., CDCA, DCA, GDCA and TDCA, bile acids have roles both in enterocytic uptake and intracellular transport of cholesterol, the latter likely “through forming bile acid/cholesterol micelles” (L39-40).

Experiments were conducted in relevant mouse models and CRL1601/NPC1L1-3xMyc-EFGP rat hepatoma cell line that expresses the liver bile acid uptake transporter Ntcp. Authors show evidence for intracellular re-localization of cholesterol and of NPC1L1 induced by the specific bile acids mentioned and demonstrate that bile acid uptake into cells is required (Ntcp KD) and FXR-independent. The authors propose that this intracellular transport function of bile acids may also be operational in hepatocytes, in which bile acid flux is related to hepatobiliary cholesterol secretion, but show no data for this assumption

The authors present a intriguing set of data and a novel hypothesis that is of interest to the field. However, there are a number of issues that require serious attention.

Fundamental concerns

1. Authors propose a mechanism by which intracellular bile acids form micelles with ERC membraneous cholesterol. This, in theory, requires the presence of (unbound) bile acids in concentrations close to their critical micellar concentrations (CMC) at the site of “solubilization”. It is generally believed that bile acids are bound to proteins during their intracellular transport (literature 1990’s). Furthermore, reported CMC values in 0.15 mM Na⁺ for the bile acids mentioned are in the order of 3-6 mM (millimolar). Cells were incubated with 1mM bile acids – data on quantification of intracellular bile acids should be provided to clarify this discrepancy. Indeed, the in vitro experiment showed transfer of liposomal cholesterol in the presence of 0.1 mM bile acids, i.e., the mechanism hereof needs to be clarified

2. The authors should discuss the (patho)physiological relevance of their finding in the light of existing knowledge about localization of cholesterol and bile acid transport along the length of the small intestine. In Figure 7F the authors provide a summarizing cartoon in which NPC1L1 and ASBT are localized in the same enterocyte. It is, however, well-established that NPC1L1 (and cholesterol absorption) is mainly localized to the jejunum while ASBT (and conjugated bile acid uptake) almost exclusively in the terminal ileum. In fact, in the jejunum, where most of cholesterol uptake is taken place, the vast majority of bile acids will be conjugated and hence not be taken up by the enterocytes. Thus, there seems to be a disconnect between the cells that transport cholesterol and those that transport bile acids in the small intestine – this issue should be discussed.

Specific issues

Line 62-63: to the best of my knowledge, there is no evidence that excess dietary cholesterol absorption causes hypercholesterolemia – should be referenced

Lines 413-416 providing explanation about bile acid concentrations used in in vitro experiments are unclear. Working concentration in water of 0.1 mmol/L, with $M \sim 400$ for bile acids would equal 0.04 mg/ml. This can not be dissolved with 50% ethanol to yield 25 mg/ml and then further diluted to working concentration

Figure 2. The authors treated NPC1L1 expressing cells with bile, dialyzed bile, boiled bile and bile acids extract. In Figure 1B, the morphology of the cells appeared to change upon treatment with 1 mM of bile acids. Would have been interesting to see a “concentration-dependent” effect. Did the authors get similar results when lower concentrations were being used and/or can the authors rule out any damaging effect on the cells when treating with bile acids. Can an effect of bile acids on the plasma membrane (including cholesterol levels in plasma membrane), result in translocation of NPC1L1?

Figure 3. The authors investigated the effect of bile acid species on CEs levels using LipidTox (stains neutral lipids, which includes CEs). CEs are often stored in lipid droplets and not only being used for VLDL production/secretion. In which compartment do the authors see the CEs accumulation. Is it ER or lipid droplets, can the authors confirm that they are lipid droplets, e.g., by performing a double staining. Do the authors see this accumulation of CEs in lipid droplets also in vivo (Figure 2E-G)?

Intracellular cholesterol transport is mediated by many different proteins and mechanisms, including NPC1/NPC2 and via the formation of membrane contact sites between organelles.

These alternative pathways cannot be excluded; therefore, the authors should discuss the possible role of other pathways in the transport of cholesterol from ERC/lysosomes to the plasma membrane/ER.

Reviewer #2 (Remarks to the Author):

In this study, Xiao et al. investigate the role of bile acids in intracellular cholesterol transport and recycling of the NPC1L1 transporter. They provide evidence for an interesting role of bile acids in cholesterol distribution between intracellular membranes. Furthermore, they show that the cholesterol-regulated translocation of the NPC1L1 transporter between endocytic recycling compartment (ERC) and plasma membrane (PM) is affected by bile acid-treatment of the used cell line. They are able to identify altered interactions of NPC1L1 with LIMA as being central for the bile-acid induced changes in NPC1L1 distribution.

Major concerns:

1. The authors use primarily filipin to assess cholesterol transport. Filipin is a polyene which binds a variety of sterols, and an important control would be to rule out, that it binds bile acids directly. In fact, filipin has been shown to bind other lipids than cholesterol, which potentially complicates the interpretation of the experiments described here. Moreover, filipin only allows for assessing the distribution of the total cellular pool of non-esterified sterols and not a particular pathway. Thus, changes in filipin staining from a perinuclear 'blob' to the PM and vice versa is no convincing evidence for sterol translocation between both compartments, as it does not rule out changes in cholesterol uptake, synthesis, or efflux. The same arguments apply to the use of D4H, which even has a higher threshold for sterol detection in membranes. In summary, additional experiments are required using alternative approaches for measuring cholesterol transport between intracellular membranes to sustain the conclusions drawn here based on filipin and D4H.

2. Quantifying the translocation of tagged NPC1L1 between ERC and PM requires an additional ERC marker, such as rab11, transferrin or others to make sure that the morphology and perinuclear location of recycling endosomes do not change during the treatments. For example, bile acids could simply cause dispersion of tubules and vesicles

belonging to the ERC, which should be ruled out.

3. Image quantification protocols are poorly documented, making it impossible to properly interpret, reconcile or reproduce many of the findings. What is meant by 'percentage of ERC cholesterol'? How precisely was this pool quantified? Similarly, how precisely was the translocation of NPC1L1 determined from acquired images? How were the 30 chosen cells selected for quantification and why so few? State-of-the art image quantification protocols allow for quantifying observations from thousands of cells per experiment and condition on standard confocal microscopes. There is a single sentence at the end of the section 'Immunofluorescence' stating that images were analyzed using ImageJ. This is far from being sufficient.

4. While the title and abstract focus on intestinal cholesterol transport, the used cell line is based on the rat hepatoma cell line McA-RH7777. These cells are not a model for enterocytes. The findings should be reproduced in a proper enterocyte cell model to justify any of the claims regarding intestinal cholesterol transport.

5. That bile acids can accelerate cholesterol exchange between membranes is an interesting hypothesis, and the developed assay can assess bile-acid mediated sterol transfer between liposomes in an elegant way. However, there are several earlier studies with similar findings, which should be cited (e.g., Vlahcevic et al. JLR, 1990, 31, 1063 and similar studies). Also, an important control would be to assess the intactness of liposomes prior and after adding BA or selected bile acids, since these are detergents, which can solubilize membranes. This could be done by dynamic light scattering, for example. Also, a question is, whether the BA extract contains bile acids in concentrations above their CMC, which would cause liposome solubilization.

6. Several claims of the manuscript are based on single experiments with small sample size (e.g., only 30 cells were analyzed in Fig. 2C, Fig. 3E and G, Fig. 4B-H, 5H-J, Fig. 6B-F and C, Fig. 7D). This is not sufficient to support the conclusions. Statistical testing should be also better described and explained, whether technical or biological replicates were included. For example, the graph in Fig. 7D is based on the mean +/- standard error of the mean from n=30 cells, likely all from the same experiment. I doubt the result of the used test given the

large error bars (especially when using SEM) and small sample size. Given the large heterogeneity of cells, much larger sample sizes and biological replicates of the experiments should be included.

Minor points:

1. The language is sometimes poor (e.g., check typos & grammar in figure legends) and imprecise, e.g.,

- p. 4, line 89: '...ERC is physically separated from the ER in intestinal enterocytes'); how has this been shown?

- p. 11, line 340: '...takes up massive macromolecules'

- figure legends could be more detailed, e.g., what means +/- cholesterol in legend to figure 1., what is & ERC cholesterol in Fig.2 and elsewhere and what means 'relative amount of mobilized cholesterol' in Fig. 3?

2. That endocytosis of NPC1L1 is essential for cholesterol absorption is likely an overstatement, given that the %reduction in cholesterol absorption is only about 50% in the knockout mice in Fig. 1. Other studies in mammalian cells also show that endocytosis of cholesterol can take place in the absence of NPC1L1. The claims made here should therefore be balanced and discussed in light of other work.

3. Do bile acids alter other lipid species than cholesterol versus cholesteryl esters?

4. Why is the ERC 'ring-shaped' in some images, for example in Fig. 6D for GDCA and TDCA in upper part?

5. While NPC1L1 is certainly an important player in intestinal cholesterol absorption, there are other transporters involved as well, e.g., SR-BI. This should be mentioned in the introduction and/or discussion eventually with percent-wise contributions, such that the reader can see the presented results in perspective.

6. The scheme in Fig. 7F can be improved, as it is very complex and unclear, what the current study contributes. Also, there is a typo in the text: change to 'bile acid transporter'.

Reviewer #3 (Remarks to the Author):

In the current manuscript, the authors propose that NPC1L1, a cholesterol transporter in the intestine that is necessary for cholesterol absorption, requires bile acids to move cholesterol within the cell to be delivered to the ER. The authors generate an interesting mouse model with a specific NPC1L1 mutation that retains NPC1L1 in the membrane and reduces cholesterol absorption in vivo. The authors then use a number of elegant in vitro studies to delineate the role of bile acids, LIMA1 in NPC1L1 cholesterol trafficking in vitro, in a hepatic cell line that stably expresses NPC1L1.

The main claim the authors make is the requirement for bile acids in the intracellular movement of cholesterol, and to this reviewer at least, that conclusion could be interpreted differently using the same data. Since bile acids are also detergents that solubilize cholesterol, and the authors have identified the better cholesterol solubilizing bile acids as key players, perhaps the results can be more simply explained. Have the authors considered that the bile acids are acting as detergents, and the results simply reflect differences in cholesterol uptake? When bile acids are present, more cholesterol enters the cell, and naturally reaches the ER. When less bile acids (or poorer detergent bile acids) are present, less cholesterol enters the cell, and is not delivered to the ER. Thus, it is critical for the authors to determine intracellular cholesterol levels in the various experiments, to rule out the role of differences in cholesterol uptake as a major factor in their conclusions. The authors can also attempt to deliver cholesterol with cyclodextrin (as opposed to take it away, as they did here), and see if the bile acids are still required. But the requirement for bile acids in the intracellular movement of cholesterol has to be resolved for the hypothesis to be verified.

The second major issue is that the title of the paper refers to intestinal cholesterol absorption, yet the authors do all the mechanistic work in a liver cell line. While I appreciate that this allows for NPC1L1 to be introduced, since the cell they utilize (rat liver cell line) does not likely have endogenous NPC1L1, the authors should carry out some experiments in an intestinal cell line to complement their in vivo model. The impressive in vivo model also appears somewhat underutilized.

The authors do not discuss the recently identified protein family (Asters) which move cholesterol from the plasma membrane to the ER. The presence of this family of proteins, which are abundant in both liver and intestine, suggest the movement of cholesterol from

plasma membrane to ER already has a specific mechanism in place. How these proteins as opposed to the putative role of bile acids, transport cholesterol intracellularly should be compared.

Lastly, the manuscript lacks important experimental details in the methods that more carefully describe how cell culture experiments are done. These are critical for the correct interpretation of the data. The statistical analysis is also not well described. Finally, some careful editing of the language is also important, although I commend the non-native speaking authors on the overall quality of the language used in the manuscript.

Reviewer #4 (Remarks to the Author):

This article by Xiao et al., reported an intracellular cholesterol transport function of bile acids and its regulation of NPC1L1 recycling. The authors proposed molecular mechanisms for the ERC-to-ER cholesterol transport mediated by bile acids and the downstream ERC-to-PM translocation of NPC1L1. To support their hypotheses, the authors used mutagenesis to assess the functional relevance of the residues participating in NPC1L1 recycling at the cellular level. This article is well written, the proposed models are novel and the data support the conclusions and claims in general. Altogether, this work is a great addition to the field. I only have one concern. As the author stated that the bile acids mediated cholesterol diffusion is nondirectional as the cholesterol levels in both ER and PM are increased. They further suggested that because the highly expressed ACAT2 converts cholesterol to cholesteryl ester, the ERC-cholesterol is mainly transported to ER. However, there is no direct data to support this conclusion. Is it possible to examine the cholesterol distribution in the ER and PM, such as using the ER and PM probes?

REVIEWER COMMENTS

Reviewer #1 (Remarks to the Author):

This interesting paper addresses (patho)physiologically important questions regarding the mechanisms that drive transport of (relatively large amounts of) absorbed cholesterol from ERC to ER inside enterocytes, thereby allowing the cholesterol to be esterified by ACAT2 for incorporation into chylomicrons or lipid droplets and, at the same time, relocation of NPC1L1 to the plasma membrane for uptake of additional cholesterol. Authors propose a novel “intracellular cholesterol transport function of bile acids” implying that specific bile acids, i.e., CDCA, DCA, GDCA and TDCA, bile acids have roles both in enterocytic uptake and intracellular transport of cholesterol, the latter likely “through forming bile acid/cholesterol micelles” (L39-40).

Experiments were conducted in relevant mouse models and CRL1601/NPC1L1-3xMyc-EFGP rat hepatoma cell line that expresses the liver bile acid uptake transporter Ntcp. Authors show evidence for intracellular re-localization of cholesterol and of NPC1L1 induced by the specific bile acids mentioned and demonstrate that bile acid uptake into cells is required (Ntcp KD) and FXR-independent. The authors propose that this intracellular transport function of bile acids may also be operational in hepatocytes, in which bile acid flux is related to hepatobiliary cholesterol secretion, but show no data for this assumption. The authors present a intriguing set of data and a novel hypothesis that is of interest to the field. However, there are a number of issues that require serious attention. **Response:** We appreciate the reviewer’s insightful and helpful comments. In the response, we have addressed all the concerns with a substantial amount of new data.

Fundamental concerns

1. Authors propose a mechanism by which intracellular bile acids form micelles with ERC membraneous cholesterol. This, in theory, requires the presence of (unbound) bile acids in concentrations close to their critical micellar concentrations (CMC) at the site of “solubilization”. It is generally believed that bile acids are bound to proteins during their intracellular transport (literature 1990’s). Furthermore, reported CMC values in 0.15 mM Na⁺ for the bile acids mentioned are in the order of 3-6 mM (millimolar). Cells were incubated with 1mM bile acids – data on quantification of intracellular bile acids should be provided to clarify this discrepancy. Indeed, the in vitro experiment showed transfer of liposomal cholesterol in the presence of 0.1 mM bile acids, i.e., the mechanism hereof needs to be clarified

Response: We thank the reviewer for this constructive suggestion.

We measured the concentrations of intracellular bile acids in the cells treated with TCA, CDCA, DCA, GDCA and TDCA (Figure-For-Reviewer 1). The concentrations in these cells are ~0.183, 0.219, 0.254, 0.182 and 0.180 mM, respectively. As pointed by the reviewer, bile acids form micelles at a high concentration and at least no less

than 2.2 mM (PMID: 33189321 and <https://doi.org/10.1007/BF01243172>). The bile acids concentration used in this study is below CMC and therefore unable to generate micelles, suggesting that the four specific bile acids at this concentration convey cholesterol independent of micelles formation.

Figure-For-Reviewer 1. Concentration of intracellular bile acids. CRL1601 cells were treated with the indicated bile acids at a concentration of 0.1mM in DMEM supplemented with 5% lipoprotein deficient serum for 2 h prior to harvesting. About 50 million cells were suspended by 2 mM EDTA in 1×PBS, rinsed by water and frozen for once. Cells were lysed by add 50% methanal and spun down. The supernatant was applied to bile acids determination. The concentrations of intracellular bile acids were calculated by the ratio of total amount of bile acids to total volume of cells. The total volume of cells is obtained by

the cell number times the average cell volume. We measured the average radius of 1000 cells using a Leica SP8 confocal microscope and obtained the average cell volume according to the volume formula of the sphere.

BAs are reported to promote cholesterol transport *in vitro* and in cultured cells at a submicellar concentration (0.15-0.6 mM) as a function of their hydrophobicity (PMIDs: 2373956, 31143889 and 9150247). Submicellar BAs can extract cholesterol from the membrane by mitigating the interaction between cholesterol and sphingomyelin, and then promote nondirectional cholesterol traffic among membranes (PMID: 9150247). To verify bile acids below their CMC could convey cholesterol, we applied the 12 bile acids to the *in vitro* cholesterol mobilizing assay. We found that all dihydroxy BAs at 0.1 mM concentration could mobilize cholesterol, while the other BAs failed to do so (New Figure 3b). In line with the previous findings (PMIDs: 2373956, 31143889), GCDCA and TCDCA at 0.1 mM concentration had less potency on cholesterol transport than CDCA, DCA, GDCA and TDCA (Figure 3b), which might account for the cholesterol entrapment in the ERC of GCDCA or TCDCA-treated cells (Figure S2d).

The six BAs (CDCA, DCA, GDCA, TDCA, GCDCA and TCDCA) transported cholesterol in a concentration-dependent manner *in vitro* (New Figure S4a). The highest concentration (1 mM) of BAs used in this experiment is still below their CMC. TCA did not mediate cholesterol transport (New Figure S4a). In addition, the dynamic light scattering assay showed that the liposomes remained intact until the BA concentration increased to 10 mM (New Figure S4b). These data indicate that the dihydroxy BAs promote cholesterol transfer directly at their submicellar

concentration without disrupting liposome structure.

The new data and discussion have been included in the revised manuscript.

2. The authors should discuss the (patho)physiological relevance of their finding in the light of existing knowledge about localization of cholesterol and bile acid transport along the length of the small intestine. In Figure 7F the authors provide a summarizing cartoon in which NPC1L1 and ASBT are localized in the same enterocyte. It is, however, well-established that NPC1L1 (and cholesterol absorption) is mainly localized to the jejunum while ASBT (and conjugated bile acid uptake) almost exclusively in the terminal ileum. In fact, in the jejunum, where most of cholesterol uptake is taken place, the vast majority of bile acids will be conjugated and hence not be taken up by the enterocytes. Thus, there seems to be a disconnect between the cells that transport cholesterol and those that transport bile acids in the small intestine – this issue should be discussed.

Response: We thank the reviewer for the constructive suggestion. We analyzed the colocalization of NPC1L1 and ASBT along the small intestine using immunoblotting. Consistent with previous reports (PMIDs: 22811412, 29880681, 16150853, 21177287), most NPC1L1 was distributed in the jejunum and ileum while ASBT is primarily localized in the ileum (New Figure S7a). Although a large amount of cholesterol absorption takes place in the jejunum, cholesterol can also be absorbed in the ileum (PMIDs: 5349613, 5429866), where NPC1L1 and ASBT are expressed.

To explore the role of ASBT in ileal cholesterol absorption, we treated mice with oral administration of linerixibat, a selective ASBT inhibitor (PMID: 23678871). Linerixibat treatment increased BA excretion by ~2-fold in feces (New Figure S7b). Administering mice with linerixibat significantly lowered total cholesterol in serum by ~26% and increased cholesterol excretion into feces by ~42% than the control (New Figure S7c-d), while the total triglyceride in serum and feces remained unaltered in the linerixibat-treated group (New Figure S7e-f). In addition, linerixibat gavage reduced cholesterol absorption by ~43% than the vehicle group (New Figure 4i). Immunohistochemical staining revealed a sequestration of NPC1L1 and cholesterol beneath the ileal brush border of the linerixibat-treated mice (New Figure 4j-k). Take together, these results suggest that ASBT is important for cholesterol absorption and NPC1L1 recycle in the ileum by mediating the entry of BAs into enterocytes.

The passive diffusion of bile acids may contribute to jejunal cholesterol absorption. It is true that ASBT is the only known intestinal bile acid transporter mediating the active transport of bile acids. However, the increased secondary bile acids ratio and the slightly decreased but not statistically significant plasma bile acids level in *Asbt* KO mice suggested that the passive absorption of bile acids or other bile acids transporters play an important role in intestinal bile acids re-absorption (PMIDs: 26022694, 36160754). Indeed, previous studies have shown that passive bile acids absorption mainly occurs in the jejunum, and the passive permeability of CDCA in the jejunum is 9-fold higher than that in the ileum (PMIDs: 4830231, 892604 and 5646181). The passive bile acids absorption in jejunum might contribute to NPC1L1-

mediated cholesterol transport.

The above results and discussion have been included in the revised manuscript.

Specific issues

Line 62-63: to the best of my knowledge, there is no evidence that excess dietary cholesterol absorption causes hypercholesterolemia – should be referenced

Response: We agree with the reviewer that it is still under debate whether dietary cholesterol cause hypercholesterolemia. However, some studies have shown higher consumption of dietary cholesterol is associated with a higher risk of CVD (PMIDs: 30874756, 33561122). We have cited these papers in the revised manuscript.

Lines 413-416 providing explanation about bile acid concentrations used in in vitro experiments are unclear. Working concentration in water of 0.1 mmol/L, with $M \sim 400$ for bile acids would equal 0.04 mg/ml. This can not be dissolved with 50% ethanol to yield 25 mg/ml and then further diluted to working concentration

Response: We apologize for not describing the method clearly. Actually, the bile acids mixture (BA) from *Sus Scrofa* was dissolved with 50% ethanol to prepare the 25 mg/mL BA stock solution. The BA mixture stock solution was filtrated with a 0.22 μm filter and diluted in DMEM with 5% lipoprotein deficient serum (LPDS) at the ratio of 1:100 to yield a working medium with 0.25 mg/mL BA mixture.

The different bile acid species, commercialized monomer compounds, were dissolved in water or DMSO and stocked in 100 mmol/L. When in use, the stock solutions of each bile acids were diluted 1000 times in DMEM with 5% LPDS and the final working concentration of these bile acids is 0.1 mmol/L.

We have rewritten the preparations of BA mixture and bile acids to make the description clearer (Pages 17-18).

Figure 2. The authors treated NPC1L1 expressing cells with bile, dialyzed bile, boiled bile and bile acids extract. In Figure 1B, the morphology of the cells appeared to change upon treatment with 1 mM of bile acids. Would have been interesting to see a “concentration-dependent” effect. Did the authors get similar results when lower concentrations were being used and/or can the authors rule out any damaging effect on the cells when treating with bile acids.

Response: As suggested by the reviewer, we treated CRL1601/NPC1L1-3xMyc-EGFP cells with varying concentrations of bile acids extract (BA) and CDX for 2 h (Figure-For-Reviewer 2). Both CDX and BA remove ERC cholesterol in a dose-dependent manner. Consistent with Figure 2b, a 2-hour treatment of CDX from 0.5% to 3% induced cell detachment and spherical shape, while any concentration of BA (0.05 – 0.5 mg/mL) did not alter cell morphology (Figure-For-Reviewer 2). In all the experiments, we used 0.25 mg/mL BA mixture to treat the cells and no obvious morphology change or cell damage was observed.

Figure-For-Reviewer 2. Cell morphology change under different concentrations of BA or CDX treatment. CRL1601/ NPC1L1-3xMyc-EGFP cells were treated with different concentrations of BA or CDX for 2 h. The cells were fixed and stained with 2 µg/mL wheat germ agglutinin (WGA) conjugates to label PM.

Can an effect of bile acids on the plasma membrane (including cholesterol levels in plasma membrane), result in translocation of NPC1L1?

Response: The 2-h BA treatment effectively increased cholesterol levels in PM (Figure 3g-h). To investigate whether the increase in PM cholesterol caused NPC1L1 translocation to PM from ERC, we treated cells with cholesterol-CDX complex (cholesterol/CDX) to increase PM cholesterol (Figure-For-Reviewer 3). Although BA and cholesterol/CDX significantly increased cholesterol levels in PM, cholesterol/CDX failed to promote the translocation of NPC1L1. Knockdown of Ntcp prevented GDCA and TDCA from entering cells and abolished NPC1L1 relocation to PM induced by these bile acid species (Figure 4a and S6b). The 10-minute BA incubation decreased ERC cholesterol and recruited LIMA1 to NPC1L1-positive ERC (Figure 6a). Together with these data, we conclude that the decrease in ERC cholesterol, but not the increase in PM, causes NPC1L1 translocation towards the PM.

Figure-For-Reviewer 3.

Cholesterol/CDX did not cause NPC1L1 to PM. CRL1601/ NPC1L1-3xMyc-EGFP cells were treated with BA mixture or cholesterol/CDX complex for 2 h. Cells were washed with 1×PBS and then stained with mcherry-D4H on ice. After incubation, cells were washed with 1×PBS and fixed with 4% PFA.

Figure 3. The authors investigated the effect of bile acid species on CEs levels using LipidTox (stains neutral lipids, which includes CEs). CEs are often stored in lipid droplets and not only being used for VLDL production/secretion. In which compartment do the authors see the CEs accumulation. Is it ER or lipid droplets, can the authors confirm that they are lipid droplets, e.g., by performing a double staining. Do the authors see this accumulation of CEs in lipid droplets also *in vivo* (Figure 2E-G)?

Response: We took the reviewer's suggestion and performed the double staining in CRL1601 cells. The CEs induced by BA or different bile acid species colocalized well with the cytosolic lipid droplet marker Plin2 (PMID: 35614132) (New Figure S4c).

We prepared intestinal organoid to mimic *in vivo* situation. We found only a portion of CEs induced by BA were colocalized with Plin2-positive lipid droplets (New Figure S4d). It is possible that the Plin2-negative lipid droplets were in ER and for lipoprotein secretion.

The above results have been included in the revised manuscript (Page 7).

Intracellular cholesterol transport is mediated by many different proteins and mechanisms, including NPC1/NPC2 and via the formation of membrane contact sites between organelles. These alternative pathways cannot be excluded; therefore, the authors should discuss the possible role of other pathways in the transport of cholesterol from ERC/lysosomes to the plasma membrane/ER.

Response: We agree with the reviewer that cholesterol is transported by different proteins and mechanisms in cells. We have taken the suggestion and discussed this point in the revised manuscript (Pages 11-12).

Reviewer #2 (Remarks to the Author):

In this study, Xiao et al. investigate the role of bile acids in intracellular cholesterol transport and recycling of the NPC1L1 transporter. They provide evidence for an interesting role of bile acids in cholesterol distribution between intracellular membranes. Furthermore, they show that the cholesterol-regulated translocation of the NPC1L1 transporter between endocytic recycling compartment (ERC) and plasma membrane (PM) is affected by bile acid-treatment of the used cell line. They are able to identify altered interactions of NPC1L1 with LIMA as being central for the bile-acid induced changes in NPC1L1 distribution.

Major concerns:

1. The authors use primarily filipin to assess cholesterol transport. Filipin is a polyene which binds a variety of sterols, and an important control would be to rule out, that it binds bile acids directly. In fact, filipin has been shown to bind other lipids than cholesterol, which potentially complicates the interpretation of the experiments described here. Moreover, filipin only allows for assessing the distribution of the total cellular pool of non-esterified sterols and not a particular pathway. Thus, changes in filipin staining from a perinuclear 'blob' to the PM and vice versa is no convincing evidence for sterol translocation between both compartments, as it does not rule out changes in cholesterol uptake, synthesis, or efflux. The same arguments apply to the use of D4H, which even has a higher threshold for sterol detection in membranes. In summary, additional experiments are required using alternative approaches for measuring cholesterol transport between intracellular membranes to sustain the conclusions drawn here based on filipin and D4H.

Response: We agree with the reviewer that filipin can bind other sterols including stigmaterol, sitosterol, campesterol and 24-methylpollinastanol besides cholesterol (PMIDs: 21412273, 18485505). However, the above sterols were not present in our system. Although filipin is not perfect, it is a well-accepted and widely used indicator of cholesterol in fixed cells.

To test whether filipin can bind different bile acid species, we depleted cellular cholesterol by 1.5% HPCD and then treated the cell with different bile acids. The cholesterol/CDX was used as a positive control. Compared with the vehicle group, the filipin signals were not increased in different bile acid treatments. However, cholesterol/CDX increased the filipin signal. Thus, filipin did not stain bile acids (Figure-For-reviewer 4).

Figure-For-Reviewer 4. Filipin does not stain bile acids. CRL1601/NPC1L1-3xMyc-EGFP cells were depleted of cholesterol by a 2-h treatment of 1.5% HPCD. Then cells were washed with 1×PBS and incubated with 2.5 μ M cholesterol/CDX, 0.25 μ g/ μ L BA mixture or 0.1 mmol/L indicated bile acid species for another 2 h. Cells were rinsed with 1×PBS, fixed by 4% PFA and stained with filipin.

As pointed out by the reviewer, D4H has a higher threshold for cholesterol recognition and the cholesterol concentration threshold required for D4H binding is about 20 mol% (PMID: 25663704). However, the plasma membrane has the highest cholesterol concentration (30-40 mol%) among all organelles, which is higher than the D4H binding threshold, and is impermeable to D4H (PMIDs: 2917977, 34118431). For these reasons, D4H was used to monitor alterations of plasma membrane cholesterol level in many studies and this study (PMIDs: 28273804, 32023146 and 31724953).

To assess cholesterol levels in ER, we analyzed the SREBP cleavage (New Figure S4f). The BA and the four bile acid species (CDCA, DCA, GDCA and TDCA) treated group had fewer mature forms of SREBP than the TCA and the control group. These results indicate that massive ERC-derived cholesterol was conveyed ER by BA or the four bile acids (CDCA, DCA, GDCA and TDCA), and then impeded SREBP cleavage. In addition, the treatment of BA or the four bile acids (CDCA, DCA, GDCA and TDCA) caused a remarkable increase in cholesteryl esters as well (New Figure 3c). These data strongly sustain our conclusion that BA or the four bile acids transport cholesterol from ERC to ER.

We agree with the reviewer that filipin and D4H are not perfect. But they are widely used in the field. With strict controls, they are powerful tools to indicate cholesterol level and distribute in the cell. Besides these two probes, we used SREBP cleavage and nascent cholesteryl ester formation (New Figure S4f, 3c-d) as surrogate markers

of cholesterol.

2. Quantifying the translocation of tagged NPC1L1 between ERC and PM requires an additional ERC marker, such as rab11, transferrin or others to make sure that the morphology and perinuclear location of recycling endosomes do not change during the treatments. For example, bile acids could simply cause dispersion of tubules and vesicles belonging to the ERC, which should be ruled out.

Response: We took the reviewer's suggestion and transfected CRL1601 cells with pCMV-mCherry-Rab11a or pCMV-TFR-cherry, respectively. BA and the four bile acid species (CDCA, DCA, GDCA and TDCA) transport cholesterol away from Rab11a or TFR-positive ERC, and could not cause the ERC redistribution (New Figure S3a-b).

The above results have been included in the revised manuscript (Page 5).

3. Image quantification protocols are poorly documented, making it impossible to properly interpret, reconcile or reproduce many of the findings. What is meant by 'percentage of ERC cholesterol'? How precisely was this pool quantified? Similarly, how precisely was the translocation of NPC1L1 determined from acquired images? How were the 30 chosen cells selected for quantification and why so few? State-of-the-art image quantification protocols allow for quantifying observations from thousands of cells per experiment and condition on standard confocal microscopes. There is a single sentence at the end of the section 'Immunofluorescence' stating that images were analyzed using ImageJ. This is far from being sufficient.

Response: We have rewritten the image quantification in the Method (Pages 22-23). 'Percentage of ERC cholesterol' is referred to the ratio of cholesterol residing in ERC to that of the whole cell. To quantify the ERC cholesterol, the cholesterol confocal images were converted to grayscale and subtracted background by the threshold tool. Then, the ERC was manually outlined according to their contours in corresponding images. The contour line, namely region of interest (ROI), was copied to the background-subtracted cholesterol image by the ROI manager tool, and the intensity of ROI in the cholesterol images was determined to obtain the intensity of ERC cholesterol. Finally, we measured the cholesterol intensity of the whole cell and calculated the percentage of ERC cholesterol by the ratio.

Similarly, the contour of ERC or the cell was outlined to generate the ROI of ERC NPC1L1 or total NPC1L1, respectively. The intensity of the two ROIs was measured and applied to ERC/total NPC1L1 calculation.

We randomly picked 30 cells per condition and used them for the quantification in the original manuscript. We have done our best to expand the quantitative size of our confocal images. We manually quantified 300 cells from 3 independent trials (100 cells per trial) for almost all the confocal images in the revised manuscript (New Figure 2c-d, 3e, 3h, 4b-c, 4f, 4h, 5b-c, 5e-f, 5h, 5j, 6b-c, 6e-f, 7e, S2e-f, S4h, S5c-d, S6d-e, S6h, S6j, S8c, S8e, S9b-c, S9 e-f, S9h and S9i). We have counted 67,800 cells from over 10,000 images, most of which were acquired in the past four months, and these updated quantitative data were consistent with our previous data. In addition, the

intensity of surface NPC1L1 was analyzed by flow cytometry, and 6,000 cells from 3 independent trials were counted (New Figure S2c, S2e and S10e). Consistently, these data from flow cytometry proved that the four bile acid species (CDCA, DCA, GDCA and TDCA) promote translocation of NPC1L1 to PM (New Figure S2c and S2g), and the A₁₂₇₂LAL truncated or mutated NPC1L1 variants are prone to translocate to PM (New Figure S10e).

4. While the title and abstract focus on intestinal cholesterol transport, the used cell line is based on the rat hepatoma cell line McA-RH7777. These cells are not a model for enterocytes. The findings should be reproduced in a proper enterocyte cell model to justify any of the claims regarding intestinal cholesterol transport.

Response: We thank the reviewer for the helpful advice.

We verified our findings in the *in vitro* cultured intestine from newborn NPC1L1-EGFP mice. NPC1L1 mainly resided on the brush border in the cholesterol-free medium (-Cholesterol group in New Figure 2e). Cholesterol gavage caused the internalization of NPC1L1 and cholesterol, which appeared as a layer below the brush border membrane (vehicle subgroup in New Figure 2e). The same pattern was observed in the group co-treated with TCA, since TCA did not mobilize intracellular cholesterol or cause NPC1L1 relocation to the PM (TCA subgroup in New Figure 2e). On the contrary, the internalized cholesterol and NPC1L1 were not seen in the intestine of mice treated with BA mixture or GDCA, both of which can mobilize ERC cholesterol and therefore cause NPC1L1 transport to the PM (BA mixture and GDCA subgroup in New Figure 2e). Meanwhile, we measured lipid droplets in the *in vitro* cultured intestine. The BA mixture and 4 specific bile acids (CDCA, DCA, GDCA and TDCA) significantly induced the production of lipid droplets, which were inhibited by avasimibe (New Figure 3f). These results are consistent with the data in cultured cells (Figure 3d) and support the notion that bile acids carry cholesterol out of the ERC and deliver cholesterol to the ER.

In sum, we have validated our key findings in the *in vitro* cultured intestine.

5. That bile acids can accelerate cholesterol exchange between membranes is an interesting hypothesis, and the developed assay can assess bile-acid mediated sterol transfer between liposomes in an elegant way. However, there are several earlier studies with similar findings, which should be cited (e.g., Vlahcevic et al. JLR, 1990, 31, 1063 and similar studies). Also, an important control would be to assess the intactness of liposomes prior and after adding BA or selected bile acids, since these are detergents, which can solubilize membranes. This could be done by dynamic light scattering, for example. Also, a question is, whether the BA extract contains bile acids in concentrations above their CMC, which would cause liposome solubilization.

Response: We thank the reviewer for these constructive suggestions. We cited these studies in our revised manuscript (Ref. 21-23 on Page 6, PMIDs: 2373956, 31143889 and 9150247).

We analyzed the diameter distribution of liposomes by dynamic light scattering (New

Figure S4b). The results showed that 10% HPCD or 10 mM bile acids can disrupt liposomes. The 3% HPCD or bile acids lower than 1 mM did change disrupt liposomes. In all experiments of this study, we treated cells with 0.1 mM bile acids, and bile acids in this concentration did not destroy the intactness of liposomes.

6. Several claims of the manuscript are based on single experiments with small sample size (e.g., only 30 cells were analyzed in Fig. 2C, Fig. 3E and G, Fig. 4B-H, 5H-J, Fig. 6B-F and C, Fig. 7D). This is not sufficient to support the conclusions. Statistical testing should be also better described and explained, whether technical or biological replicates were included. For example, the graph in Fig. 7D is based on the mean +/- standard error of the mean from n=30 cells, likely all from the same experiment. I doubt the result of the used test given the large error bars (especially when using SEM) and small sample size. Given the large heterogeneity of cells, much larger sample sizes and biological replicates of the experiments should be included.

Response: We took the reviewer's suggestion and manually quantified 300 cells from 3 independent trials for almost all the confocal images, including the graphs in Figure 7d, in the revised manuscript (New Figure 2c-d, 3e, 3h, 4b-c, 4f, 4h, 5b-c, 5e-f, 5h, 5j, 6b-c, 6e-f, 7e, S2e-f, S4h, S5c-d, S6d-e, S6h, S6j, S8c, S8e, S9b-c, S9 e-f, S9h and S9i).

To make quantifications of Figure 7d more credible, we constructed CRL1601/NPC1L1-3xMyc-EGFP- ΔA_{1272} LAL or CRL1601/NPC1L1-3xMyc-EGFP- A_{1272} LAL \rightarrow 4E stable cell lines for surface NPC1L1 quantification by flow cytometry analysis (New Figure S10e), and 6,000 cells from 3 independent trials (2000 cells/trial) were analyzed. Consistent with our previous statistical data, these quantitative data sustained our conclusions.

Minor points:

1. The language is sometimes poor (e.g., check typos & grammar in figure legends) and imprecise, e.g.,

Response: We thank the reviewer for the helpful advice. Several sentences have been rephrased to make our expression clearer.

- p. 4, line 89: '...ERC is physically separated from the ER in intestinal enterocytes'); how has this been shown?

Response: The sentence has been revised to 'However, the immunohistochemical staining results showed that the Rab11-positive ERC displayed little overlap with the ACAT2-positive ER in intestinal enterocytes' on Page 3.

- p. 11, line 340: '...takes up massive macromolecules'

Response: The sentence has been revised to "Extensive apical endocytic membrane structures are observed in the neonatal rat and the pup absorbs dietary fat through endocytosis'.

- figure legends could be more detailed, e.g., what means +/- cholesterol in legend to figure 1., what is & ERC cholesterol in Fig.2 and elsewhere and what means 'relative

amount of mobilized cholesterol' in Fig. 3?

Response: We thank the reviewer for reading carefully and have made corrections according to the reviewer's comments.

' \pm cholesterol' in Figure 1b means oral gavage with 100 μ L corn oil in the presence (+ cholesterol group) or absence (- cholesterol group) of 30 mg/mL cholesterol.

'The ERC cholesterol' in Figure 2c was defined as the cholesterol colocalized with intracellular NPC1L1

'Relative amount of mobilized cholesterol' is the specific value that the total cholesterol mobilized to the non-cholesterol liposome divided by the total cholesterol in the cholesterol-liposome.

These corrections have been described in the Legends of Figure 1-3 and Methods.

2. That endocytosis of NPC1L1 is essential for cholesterol absorption is likely an overstatement, given that the %reduction in cholesterol absorption is only about 50% in the knockout mice in Fig. 1. Other studies in mammalian cells also show that endocytosis of cholesterol can take place in the absence of NPC1L1. The claims made here should therefore be balanced and discussed in light of other work.

Response: We agree with the reviewer that there might be other proteins to mediate cholesterol absorption besides NPC1L1, such as scavenger receptor class B type I (SR-BI) or ASTERs. We discussed this point in the revised manuscript (Pages 11-12).

3. Do bile acids alter other lipid species than cholesterol versus cholesteryl esters?

Response: We tested whether the 2-h treatment of BA changed PI(4,5)P₂ (probed by PH-PLC δ -mCherry) or PI3P (EGFP-FYVE_{SARA}) distribution in CRL1601 cells (New Figure S3c-d). BA effectively decreased cholesterol in the ERC but did not alter PI(4,5)P₂ or PI3P distribution.

4. Why is the ERC 'ring-shaped' in some images, for example in Fig. 6D for GDCA and TDCA in upper part?

Response: We only detected the ERC 'ring-shaped' in some LIMA1 images (Figure 6d). LIMA1 is localized in the pericellular region, including peripheral actin and PM (PMIDs: 24694988 and 29880681), while NPC1L1 accumulates at ERC. When ERC cholesterol decreases, NPC1L1 of ERC recruits LIMA1 from the pericellular region. Considering the short-time recruitment (10 minutes), it is reasonable to see the ring-shaped LIMA1 in some images as LIMA1 binds to the cytosolic tail of NPC1L1.

5. While NPC1L1 is certainly an important player in intestinal cholesterol absorption, there are other transporters involved as well, e.g., SR-BI. This should be mentioned in the introduction and/or discussion eventually with percent-wise contributions, such that the reader can see the presented results in perspective.

Response: We took the suggestion and discussed the role and contribution of SR-BI in intestinal cholesterol absorption in the discussion (Pages 11-12).

6. The scheme in Fig. 7F can be improved, as it is very complex and unclear, what the current study contributes. Also, there is a typo in the text: change to 'bile acid transporter'.

Response: As shown in Figure 7f, we simplified and redrawn our scheme according to the suggestion. The typos have been corrected in the revised manuscript.

Reviewer #3 (Remarks to the Author):

In the current manuscript, the authors propose that NPC1L1, a cholesterol transporter in the intestine that is necessary for cholesterol absorption, requires bile acids to move cholesterol within the cell to be delivered to the ER. The authors generate an interesting mouse model with a specific NPC1L1 mutation that retains NPC1L1 in the membrane and reduces cholesterol absorption in vivo. The authors then use a number of elegant in vitro studies to delineate the role of bile acids, LIMA1 in NPC1L1 cholesterol trafficking in vitro, in a hepatic cell line that stably expresses NPC1L1.

Response: We appreciate the reviewer's positive comments and helpful suggestions.

The main claim the authors make is the requirement for bile acids in the intracellular movement of cholesterol, and to this reviewer at least, that conclusion could be interpreted differently using the same data. Since bile acids are also detergents that solubilize cholesterol, and the authors have identified the better cholesterol solubilizing bile acids as key players, perhaps the results can be more simply explained. Have the authors considered that the bile acids are acting as detergents, and the results simply reflect differences in cholesterol uptake? When bile acids are present, more cholesterol enters the cell, and naturally reaches the ER. When less bile acids (or poorer detergent bile acids) are present, less cholesterol enters the cell, and is not delivered to the ER. Thus, it is critical for the authors to determine intracellular cholesterol levels in the various experiments, to rule out the role of differences in cholesterol uptake as a major factor in their conclusions. The authors can also attempt to deliver cholesterol with cyclodextrin (as opposed to take it away, as they did here), and see if the bile acids are still required. But the requirement for bile acids in the intracellular movement of cholesterol has to be resolved for the hypothesis to be verified.

Response: In this study, we incubated CRL1601/NPC1L1-3xMyc-EGFP with BA mixture or specific bile acid species in DMEM containing 5% lipoprotein deficient serum (LPDS). To make the experiment more rigorous, the cells were incubated with BA in the medium DMEM/1 μ M lovastatin with or w/o 5% LPDS, in which de novo cholesterol synthesis was blocked and lacked any exogenous cholesterol. BA could still transport ERC cholesterol away and translocate ERC NPC1L1 to PM (Figure-For-Reviewer 5).

Figure-For-Reviewer 5: BA-stimulated transport of intracellular cholesterol and NPC1L1 is independent of exogenous cholesterol. CRL1601/ NPC1L1-3xMyc-EGFP cells were treated with BA mixture in DMEM/1 μ M lovastatin in the presence or absence of 5% lipoprotein deficient serum for 2 h. Cells were fixed and stained with 50 μ g/mL filipin.

We took the reviewer's suggestion and examined whether BA promoted the movement of ERC cholesterol in the presence of cholesterol/CDX complex. We found that BA decreased ERC cholesterol regardless of cholesterol/CDX treatment (Figure-For-Reviewer 6a). Consistently, the four specific bile acids (CDCA, DCA, GDCA and TDCA) could transport cholesterol out of ERC even the cells were incubated with cholesterol/CDX (Figure-For-Reviewer 6b).

NTCP transports GDCA and TDCA, but not CDCA or DCA into cells (Figure S6b). Even with cholesterol/cdx, knockdown of *Ntcp* blocked cholesterol egress from ERC mediated by GDCA or TDCA, but not by CDCA or DCA (Figure-For-Reviewer 6b), indicating bile acids act in the cells and do not reflect cholesterol influx.

Figure-For-Reviewer 6: Exogenous cholesterol is not required for bile acids-induced ERC cholesterol egress. a CRL1601/ NPC1L1-3xMyc-EGFP cells were treated with BA mixture in DMEM/5% lipoprotein deficient serum in the presence or absence of 1

μ M cholesterol/CDX complex. Cells were fixed and stained with 50 μ g/mL filipin. **b** CRL1601/ NPC1L1-3xMyc-EGFP cells were transfected with scramble siRNA or siRNA against rat *Ntcp* for 48 h. cells were treated with indicated bile acids for 2 h in the presence of 1 μ M cholesterol/CDX. Cells were fixed and stained with 50 μ g/mL filipin.

In summary, we demonstrate that bile acids transport ERC cholesterol away by acting as a cholesterol mobilizer in the cell. BA did so independent of exogenous cholesterol because: firstly, no exogenous cholesterol was available during bile acids incubation; Secondly, ERC cholesterol was indeed transported away after the incubation of bile acids (Figure 2b, 5a); Lastly, even with exogenous cholesterol bile acids still promoted the egress of ERC cholesterol, and the process was regulated by whether bile acids entered into cells rather than the presence of exogenous cholesterol (Figure-For-Reviewer 6).

The second major issue is that the title of the paper refers to intestinal cholesterol absorption, yet the authors do all the mechanistic work in a liver cell line. While I appreciate that this allows for NPC1L1 to be introduced, since the cell they utilize (rat liver cell line) does not likely have endogenous NPC1L1, the authors should carry out some experiments in an intestinal cell line to complement their in vivo model. The impressive in vivo model also appears somewhat underutilized.

Response: We thank the reviewer for this insightful suggestion.

Two key experiments were performed in enterocytes from *in vitro* 3D cultured intestines to verify bile acids participate in NPC1L1-mediated cholesterol transport within cells. We checked whether BA transported cholesterol away from endocytic NPC1L1 vesicles. In new Figure 2e, cholesterol incubation induced endocytosis of NPC1L1 together with cholesterol. BA conveyed cholesterol out of the NPC1L1-positive subapical layer and relocated the endocytic NPC1L1 to the brush border, while TCA failed to do so. We also conducted the LipidTox staining assay in the intestine after indicated bile acids incubation with or w/o avasimibe. BA and 4 specific bile acids (CDCA, DCA, GDCA and TDCA) significantly induced the production of lipid droplets, which could be inhibited by avasimibe treatment (New Figure 3f). TCA did not alter lipid droplets. The results from the *in vitro* cultured intestine were consistent with these results in CRL1601/NPC1L1-3xMyc-EGFP cells and support our notion that bile acids convey cholesterol from NPC1L1-positive vesicles to ER.

In addition, we treated mice with oral administration of linerixibat, a selective ASBT inhibitor (PMID: 23678871). Linerixibat treatment increased bile acid excretion by ~2-fold in feces from mice (New Figure S7b), indicating the compound successfully prevents ileum from re-absorbing bile acids. Linerixibat gavage reduced cholesterol absorption of mice by ~43% (New Figure 4i). Immunohistochemical staining revealed that substantial NPC1L1 and cholesterol were sequestered beneath the ileal brush border of the linerixibat-treated mice, while no subapical NPC1L1 or cholesterol was found in the intestine from the vehicle group (New Figure 4j-k). Overall, these results suggested that bile acids are transported into enterocytes, mobilize the ERC-

cholesterol and then induce NPC1L1 translocation.

We have included these results and discussions in our revised manuscript.

The authors do not discuss the recently identified protein family (Asters) which move cholesterol from the plasma membrane to the ER. The presence of this family of proteins, which are abundant in both liver and intestine, suggest the movement of cholesterol from plasma membrane to ER already has a specific mechanism in place. How these proteins as opposed to the putative role of bile acids, transport cholesterol intracellularly should be compared.

Response: We agree with the reviewer that Asters might play a role in intestinal cholesterol absorption. We have discussed this point in the revised manuscript (Page 12).

Lastly, the manuscript lacks important experimental details in the methods that more carefully describe how cell culture experiments are done. These are critical for the correct interpretation of the data. The statistical analysis is also not well described. Finally, some careful editing of the language is also important, although I commend the non-native speaking authors on the overall quality of the language used in the manuscript.

Response: We have substantially revised the **Method** section and added experimental details (Pages 14-25). We clearly described how each type of data was analyzed in the **Statistical analysis** section (Page 25).

Reviewer #4 (Remarks to the Author):

This article by Xiao et al., reported an intracellular cholesterol transport function of bile acids and its regulation of NPC1L1 recycling. The authors proposed molecular mechanisms for the ERC-to-ER cholesterol transport mediated by bile acids and the downstream ERC-to-PM translocation of NPC1L1. To support their hypotheses, the authors used mutagenesis to assess the functional relevance of the residues participating in NPC1L1 recycling at the cellular level. This article is well written, the proposed models are novel and the data support the conclusions and claims in general. Altogether, this work is a great addition to the field. I only have one concern. As the author stated that the bile acids mediated cholesterol diffusion is nondirectional as the cholesterol levels in both ER and PM are increased. They further suggested that because the highly expressed ACAT2 converts cholesterol to cholesteryl ester, the ERC-cholesterol is mainly transported to ER. However, there is no direct data to support this conclusion. Is it possible to examine the cholesterol distribution in the ER and PM, such as using the ER and PM probes?

Response: We are grateful to review #4 for the useful suggestion and positive feedback.

To date, there are only two probes, D4H and filipin for directly labeling cholesterol. Filipin is suitable for imaging fixative cells owing to its toxicity, while D4H is often

used to indicate organelles with high cholesterol level because of its higher threshold for cholesterol detection. ER cholesterol cannot be directly probed as cholesterol concentration in ER, occupying 5% of total ER lipids (PMID: 19041766), is too low to detect.

We therefore apply SREBP2 cleavage analysis to indirectly indicate the transport of ERC cholesterol to ER. The SREBP2 cleavage activation depends on the ER cholesterol concentration, which exceeds 5 mol% of total ER lipids block SREBP2 cleavage (PMID: 19041766). The cholesterol starvation did not change ERC cholesterol levels (New Figure S4e). The BA mixture and the four specific bile acids effectively inhibited the SREBP2 processing, and TCA failed to do so (New Figure S4f). These data suggest that bile acids directly convey cholesterol from ERC to ER.

Overexpressing ACAT2 was unable to reduce cholesterol levels in PM or increase the amount of lipid droplets (Figure-For-Reviewer 7a-b). We speculate that ACATs were highly active and sufficiently esterified ER cholesterol. Next, we treated cells with ACATs inhibitor avasimibe during bile acids treatment. Avasimibe largely increased the cholesterol level of the plasma membrane in BA mixture, CDCA, DCA, GDCA or TDCA-treated groups, respectively (New Figure S4g-h). In contrast, avasimibe had little or no effects on the cholesterol level of PM in the vehicle or TCA-treated group. These data suggest that the ACAT activity can alter the destination of cholesterol from ERC.

Figure-For-Reviewer 7: Overexpression of ACAT2 has no effects on bile acids-induced increment of lipid droplets or PM cholesterol. a-b CRL1601 cells were transfected with pcDNA3 or pCMV-ACAT2-3xFLAG for 48 h and then treated with indicated bile acids for 2 h prior to harvesting. Cells were stained with D4H, fixed by 4% PFA and incubated with an anti-FLAG antibody (For a). As to Bodipy staining (to label lipid droplets), cells were fixed, stained with anti-FLAG antibody and then

counterstained with Bodipy.

REVIEWERS' COMMENTS

Reviewer #1 (Remarks to the Author):

The authors have done extensive additional experiments and have adapted the text at several locations, which has strengthened the manuscript. Importantly, it is now explicitly stated that intracellular bile acids must act in a non-micellar fashion, which indeed is supported by their own data. There is one issue that does need attention: the authors address the discrepancy between the majority of cholesterol absorption in the upper small intestine and ASBT-mediated bile acid uptake in the ileum in their reaction to the reviewers, yet, in the discussion (mid paragraph page 19) this is not at all addressed. In addition, the cartoon in Figure 7f has remained unchanged, implicitly stating that ASBT-mediated bile acid uptake is linked to cholesterol absorption while, in reality, passive bile acid absorption in the upper intestine will be much more important in the context of this work. This should be adapted, both in text and in Figure.

Small point: there are several small typos in the new text parts.

Reviewer #2 (Remarks to the Author):

I think, the authors have answered the majority of my questions. Importantly, they have carried out a large number of additional experiments, controls and quantifications to meet the raised concerns.

With these changes, the manuscript can be accepted for publication.

Reviewer #3 (Remarks to the Author):

The authors have done a good job with the revisions, and I have no issues with the manuscript.

Reviewer #4 (Remarks to the Author):

The authors have done an excellent job in addressing my concerns. I support the publication of this manuscript.

REVIEWERS' COMMENTS

Reviewer #1 (Remarks to the Author):

The authors have done extensive additional experiments and have adapted the text at several location, which has strengthened the manuscript. Importantly, it is now explicitly stated that intracellular bile acids must act in a non-micellar fashion, which indeed is supported by their own data.

Response: We appreciate this reviewer for the positive comments.

There is one issue that does need attention: the authors address the discrepancy between the majority of cholesterol absorption in the upper small intestine and ASBT-mediated bile acid uptake in the ileum in their reaction to the reviewers, yet, in the discussion (mid paragraph page 19) this is not at all addressed. In addition, the cartoon in Figure 7f has remained unchanged, implicitly stating that ASBT-mediated bile acid uptake is linked to cholesterol absorption while, in reality, passive bile acid absorption in the upper intestine will be much more important in the context of this work. This should be adapted, both in text and in Figure.

Response: We agree with the reviewer that the upper small intestine is responsible for the majority of cholesterol absorption. We took the advice and removed ASBT/BA transporter from the cartoon in Figure 7f. The sentence in the discussion has been revised as 'BAs can be imported by passive diffusion in the duodenum and jejunum or by active transport involving ASBT in the ileum. The passive diffusion of bile acids might play a more important role in cholesterol absorption, since a large amount of cholesterol absorption takes place in the upper intestine'.

Small point: there are several small typo's in the new text parts.

Response: We thank the reviewer for the kind reminder. We have carefully proofread the manuscript and corrected the typos.

Reviewer #2 (Remarks to the Author):

I think, the authors have answered the majority of my questions. Importantly, they have carried out a large number of additional experiments, controls and quantifications to meet the raised concerns.

With these changes, the manuscript can be accepted for publication.

Response: We thank this reviewer for his/her valuable comments and helpful suggestions for improving the manuscript.

Reviewer #3 (Remarks to the Author):

The authors have done a good job with the revisions, and I have no issues with the manuscript.

Response: We thank the reviewer for helping us to improve our manuscript.

Reviewer #4 (Remarks to the Author):

The authors have done an excellent job in addressing my concerns. I support the publication of this manuscript.

Response: We greatly appreciate the reviewer for the positive comments.